# SNAPHARD CONTRAST LEARNING

**Changpu Meng[1], Jie Yang[1],\* Wanqing Li[1], Yi Guo[2]\***
[1]University of Wollongong
[2]Western Sydney University
`cm245@uowmail.edu.au,jiey@uow.edu.au,wanqing@uow.edu.au,`
`guo@westernsydney.edu.au`

## ABSTRACT

In recent years, Contrastive Learning (CL) has garnered significant attention due to its efficacy across various domains, spanning from visual and textual modalities. A fundamental aspect of CL is aligning the representations of anchor instances with relevant positive samples while simultaneously separating them from negative ones. Prior studies have extensively explored diverse strategies for generating and sampling contrastive (*i.e.*, positive/negative) pairs. Despite the empirical success, the theoretical understanding of the CL approach remains under-explored, leaving questions such as the rationale behind contrastive-pair sampling and its contributions to the model performance unclear. This paper addresses this gap by providing a comprehensive theoretical analysis from the angle of optimality conditions and introducing the SnaPhArd Contrast Learning (*SPACL*). Specifically, *SPACL* prioritizes hard positive and hard negative samples during constructing contrastive pairs and computing the contrastive loss, rather than treating all samples equally. Experimental results across three downstream tasks demonstrate that *SPACL* consistently outperforms or competes favorably with state-of-the-art methods, showcasing its robustness and efficacy. A comprehensive ablation study further examines the effectiveness of *SPACL*'s individual components to verify the theoretic findings.

## 1 INTRODUCTION

Contrastive Learning (CL) has proven to be a powerful self-supervised approach for reducing the reliance on large volumes of labeled data in training deep neural networks. At its essence, CL harnesses self-augmented signals from input data and revolves via optimizing a contrastive loss function. Throughout training, the model generates discriminative features, bringing similar (or positive) samples closer in the representation space while pushing dissimilar (or negative) samples apart. Thanks to these discriminative features, CL has led to notable advancements in various domains, including image classification (Mildenberger et al., 2025), relation extraction (Wu et al., 2023), and visual question answering (Zou & Yin, 2025), to name a few.

The effectiveness of CL is predominantly tied to the strategic utilization of contrastive pairs (both positive and negative examples) to discover informative features. Consequently, extensive investigations have been conducted, resulting in the identification of two primary categories within the literature. One category focuses explicitly on the **generation of contrastive pairs**. Techniques such as *minibatch adaptation*, *data augmentation*, and *knowledge-based* methods have been explored to enhance the quality and diversity of these contrastive pairs. Among them, minibatch adaptation, exemplified by the MoCo series and SimCLR, leverages in-batch data and/or a global queue to retrieve contrastive samples (Koromilas et al., 2024; Lin et al., 2024). Data augmentation techniques introduce variability and richness to the data, employing transformations at both instance and feature levels (Liu et al., 2024c; Zheng et al., 2024; Liu et al., 2024c; Zhuo et al., 2024; Lei et al., 2024). Knowledge-based methods, on the other hand, integrate domain knowledge and/or prior information to generate contrastive pairs, especially in applications such as social recommendation and machine reading comprehension (Animesh & Chandraker, 2025; Lin et al., 2024; Li et al., 2024b; Lee et al., 2024b; Kim et al., 2024). Alternatively, another category values the **sampling of contrastive pairs**,

---
*Corresponding authors.

with a growing emphasis on utilizing hard (positive and negative) samples (Zhuang et al., 2024a; Zou & Yin, 2025; Zhang et al., 2024). The sample hardness or difficulty refers to instances that present greater challenges or ambiguity for the model in differentiating them from counter class(es). The integration of such hard samples prompts the model discriminatory abilities via capturing subtle variations within the data. Accordingly, diverse strategies have been proposed to proficiently sample contrastive pairs, including feature importance (Chin et al., 2024), model uncertainty (Liu et al., 2025a), and data bias (Li et al., 2024c), among others.

Although generating contrastive pairs is crucial in CL, our emphasis lies in the second category, namely the strategic sampling of contrastive pairs. Despite the empirical success of pair sampling, a gap persists in the theoretical understanding. Specifically, questions such as *the motivation or rationale behind sampling contrastive pairs*, *the contribution of hard samples to the model performance*, and *the conditions for solution optimality/collapse under the circumstances of hard samples* remain open. Moreover, most existing research (Shou et al., 2025; Wang et al., 2024a; Wu et al., 2025a) focuses solely on sampling either positive or negative samples, with fewer considering both simultaneously. Consequently, the *selection of both positive and negative samples concurrently* introduces another complex perspective. Our work in this paper provides a response to these issues, which is summarized as below:

- A theoretical analysis on current Contrastive Learning methods, outlining conditions for optimality, and exploring potential issues such as solution collapse.
- Based on the analysis, a screening-based Contrastive Learning algorithm is proposed to explicitly harness hard contrastive pairs.
- Empirical evaluation is performed on three representative downstream tasks, namely image classification, Knowledge-Graph link prediction and out-of-domain detection, to comprehensively assess the effectiveness of *SPACL*. Furthermore, an extensive ablation study is carried out to examine the contribution of each individual component.

## 2 DIVE DEEP INTO CONTRASTIVE LEARNING

### 2.1 CONTRASTIVE LEARNING FORMULATION

A large body of Contrastive Learning (CL) methods adopt a dual-encoder architecture, where two encoders separately encode an anchor (query) and its associated positive (key) sample. This general structure underlies several well-known instantiations, such as Momentum Contrastive Learning (He et al., 2020; Chen et al., 2020b; Zhu et al., 2021; Liu et al., 2024a; Chen et al., 2024; Wu et al., 2025b)[1]. Let $\mathbf{z}_{i_j}^-/\mathbf{z}_{i_j}^+$ be the $j$-th negative/positive sample of an anchor sample $\mathbf{z}_i$, and $N^-/N^+$ be the total number of negative/positive samples for $\mathbf{z}_i$. With a similarity measure by the dot product (*i.e.*, $\langle \cdot, \cdot \rangle$) and a hyperparameter $\tau$ as the temperature, the following InfoNCE loss is applied,

$$\Theta^* = \arg\min_{\Theta} - \sum_i \log \left\{ \frac{\sum_j^{N^+} \exp(\langle f(\mathbf{z}_i), g(\mathbf{z}_{i_j}^+) \rangle / \tau)}{\sum_j^{N^+} \exp(\langle f(\mathbf{z}_i), g(\mathbf{z}_{i_j}^+) \rangle / \tau) + \sum_j^{N^-} \exp(\langle f(\mathbf{z}_i), g(\mathbf{z}_{i_j}^-) \rangle / \tau)} \right\}, \quad (1)$$

to optimize the *main* encoder $f_\Theta(\mathbf{z}_i)$ with model parameters of $\Theta$ (omitting $\Theta$ in $f$ for ease of notation). In Eq. (1), $g$ is the *derivative* encoder (often omitted during inference) and is often intentionally aligned with $f$, albeit updated differently. This alignment may involve incorporating a (very) small portion of the updated $\Theta$ from $f$ in a momentum-based manner while maintaining other portions. This InfoNCE loss tends to be minimal when $f(\mathbf{z}_i)$ closely matches its positive counterparts and diverges from negatives.

### 2.2 OPTIMIZATION

Write $\mathbf{x}_i = f(\mathbf{z}_i)$, $\mathbf{n}_j = g(\mathbf{z}_{i_j}^-)$ and $\mathbf{p}_j = g(\mathbf{z}_{i_j}^+)$. We first note that the function of the temperature $\tau$ is to change the weighting scheme and does not affect our theoretic results, as verified in the

---

[1]Unless otherwise specified, our analysis concerns dual-encoder based CL, where two branches produce representations for a query–key pair (parameters may be shared or partially decoupled via momentum). Methods that use a single backbone with asymmetric heads and stop-gradient mechanisms (*e.g.*, BYOL (Grill et al., 2020), SimSiam (Chen & He, 2021)) can often be mapped to the dual-encoder template by viewing the predictor/teacher head as an effective second encoder. When such a mapping is not appropriate, we explicitly note it in the text.

derivations. Hence we omit $\tau$ in our following analysis for simplicity. We further write $s_i^+ := \sum_j^{N^+} \exp\langle \mathbf{x}_i, \mathbf{p}_j \rangle$ and $s_i^- := \sum_j^{N^-} \exp\langle \mathbf{x}_i, \mathbf{n}_j \rangle$. Then the objective from Eq. (1) is concisely to minimize the below $w.r.t$ $\Theta$:

$$\mathcal{L} := -\sum_i \log \left\{ \frac{\sum_j^{N^+} \exp\langle \mathbf{x}_i, \mathbf{p}_j \rangle}{\sum_j^{N^+} \exp\langle \mathbf{x}_i, \mathbf{p}_j \rangle + \sum_j^{N^-} \exp\langle \mathbf{x}_i, \mathbf{n}_j \rangle} \right\} = \sum_i \log \left\{ \frac{s_i^-}{s_i^+} + 1 \right\}. \quad (2)$$

We point out that although $\mathbf{n}_j$ and $\mathbf{p}_j$ are encoded by the derivative encoder $g$, it is constant during the optimization of $f$, and so are $\mathbf{n}_j$ and $\mathbf{p}_j$ $w.r.t$ $\mathbf{x}_i$. Accordingly, the gradient $w.r.t$ $\Theta$ is

$$\frac{\partial \mathcal{L}}{\partial \Theta} = \sum_i \left\{ \frac{1}{s_i^-/s_i^+ + 1} \frac{\partial}{\partial \mathbf{x}_i} \left( \frac{s_i^-}{s_i^+} \right) \otimes \frac{\partial \mathbf{x}_i}{\partial \Theta} \right\}, \quad (3)$$

where $\otimes$ is some tensor product derived from chain rule. In addition, $f$ is typically a deep neural network with considerable complexity. The universal approximation property of neural networks implies the existence of $\Theta$ for arbitrary mappings between $\mathbf{z}$ and $\mathbf{x}$ such that $\mathbf{x} = f(\mathbf{z})$. Therefore, the optimization process can be streamlined by prioritizing the optimal values for $\mathbf{x}_i$. This involves first identifying the best $\mathbf{x}_i$, and then pursuing them with an appropriate $\Theta$ as a regression problem. The motivation lies in the geometry of the unit hypersphere. That is, when $\mathbf{n}_j$ and $\mathbf{p}_j$ remain constant, the optimal $\mathbf{x}_i$ corresponding to the minimum of $\mathcal{L}$ becomes predictable, or at least some of its properties are tangible. As a result, the stationary point for Eq. (3) is (hereafter omitting $i$ for simplicity) is given by: $\partial (s^-/s^+)/\partial \mathbf{x} = 0$. This condition is further equivalent to the following constrained minimization problem:

$$\min_{\|\mathbf{x}\|_2^2 = 1} \frac{\sum_j \exp\langle \mathbf{x}, \mathbf{n}_j \rangle}{\sum_j \exp\langle \mathbf{x}, \mathbf{p}_j \rangle}, \quad (4)$$

or, alternatively, an explicit minimization problem (without constraints):

$$\min_{\mathbf{x}} \frac{\sum_j \exp\langle \frac{\mathbf{x}}{\|\mathbf{x}\|}, \mathbf{n}_j \rangle}{\sum_j \exp\langle \frac{\mathbf{x}}{\|\mathbf{x}\|}, \mathbf{p}_j \rangle}. \quad (5)$$

We begin with the constrained version, offering a direct geometric interpretation, before showing that the unconstrained version closely resembles its constrained counterpart.

### 2.2.1 Optimality condition of the constrained version

We first provide the following theorem to establish the condition for solution collapsing with *a single positive sample*. By "collapsing", we refer to the scenario where $\mathbf{x} = \mathbf{p}$, indicating that the optimal solution coincides with the positive sample itself.

**Theorem 2.1** (Symmetric negatives). *If there is only a single positive sample* $\mathbf{p}$, *i.e.,* $N^+ = 1$, *the optimal solution* $\mathbf{x}$ *will collapse to the only positive one* $\mathbf{p}$ *if and only if the following holds with some* $C \in \mathbb{R}$

$$\frac{\sum_j w_j \mathbf{n}_j}{C} = \mathbf{p}, \ \forall j, \ w_j = \frac{\exp\langle \mathbf{n}_j, \mathbf{p} \rangle}{\sum_{j=1}^{N^-} \exp\langle \mathbf{n}_j, \mathbf{p} \rangle}. \quad (6)$$

The detailed proof and illustrations are in A.1.1. The symmetry condition in **Theorem** 2.1 indicates that *when the convex reconstruction of negatives is a scaled version of the single positive, the optimal solution collapses*. It also implies that collapsing does not occur without symmetry. Consequently, we derive the following non-collapsing condition as a corollary.

**Corollary 2.2** (Non-collapsing condition for a single positive). *For the same setting as in* **Theorem** *2.1, if* $\forall c \in \mathbb{R} \backslash \{0\}$, $c\mathbf{p} \notin conv\{\mathbf{n}_j\}_{j=1}^{N^-}$, *then* $\mathbf{x} \neq \mathbf{p}$. *conv{} is given in Appendix Eq.* (12).

*Proof.* This condition is a relaxed version of **Theorem** 2.1, as Eq. (6) explicitly demands the convex reconstruction. When this condition, as stated in this corollary, is not met, it becomes impossible to have $\mathbf{x} = \mathbf{p}$. □

**Remark 2.3.** *The condition outlined in **Corollary** 2.2 can also be conceptualized as the absence of intersection between the projective space of $\mathbf{p}$ (the line connecting $\mathbf{p}$ and $-\mathbf{p}$) and the convex body formed by all negatives. This condition holds potential when all the negatives collectively occupy less than half of the sphere, for instance, concentrating around a small sphere cap. However, if the negatives encompass a substantial portion of the sphere, the condition becomes less informative, and one must refer to Eq. (6) to assess feasibility.*

A related geometric visualization provided in Fig. 7 (Appendix) offers a clear depiction of the interactions among instances. First, *the coverage of the negatives is primarily determined by the presence of hard samples*. The easy ones are mostly concentrated around the opposite side of $f(\mathbf{z})$ or $\mathbf{x}$. The more of these easy ones, the more weights they attract, which is a "fixing" force for the potential solution. In other words, $\mathbf{n}(\mathbf{x})$ would be closer to them and hence more difficult for the solution to break free from them. This suggests to filter out some easy negatives. The second is to apply the same logic to the positives. The easy positive instances that are close to $f(\mathbf{z})$ also serve as fixation points, expanding the convex hull for $\mathbf{p}(\mathbf{x})$. The consequence is not only discouraging variations in the encoder, but also potentially yielding multiple ambiguous solutions. Therefore, we should eliminate the easy samples, both negatives and positives, to alleviate these potential issues.

Interestingly, when $\|\mathbf{x}\|$ is constrained to be on $\mathcal{S}^{m-1}$, we have the following asymptotic results on the number of negative instances. Proof and illustrations are in A.1.2.

**Lemma 2.4.** *When $\mathbf{x} \in \mathcal{S}^{m-1}$, i.e. $\|\mathbf{x}\| = 1$, the norm of the gradient is upper-bounded asymptotically, i.e.*

$$\lim_{N^- \to \infty} \|\nabla_{\mathbf{x}} \mathcal{L}_c\| \leq 2.$$

**Remark 2.5.** *The proof of **Lemma** 2.4 and its illustration in Fig. 8 (Appendix) clearly demonstrate that the length of the gradient $\nabla_{\mathbf{x}} \mathcal{L}_c$ is capped by the constant 2. Consequently, having an excessive number of negatives becomes unnecessary. That is, when $N^-$ surpasses a critical threshold, the contribution of negatives becomes negligible. In addition, from Eq. (15) we observe that*

$$\|\nabla_{\mathbf{x}} \mathcal{L}_c\| \leq \frac{2}{e^{-2} + 1} \approx 1.76.$$

*This represents the maximum gradient length when $N^- = 1$ and $N^+ = 1$. As $N^-$ increases, $\|\nabla_{\mathbf{x}} \mathcal{L}_c\|$ experiences a slight increase and rapidly approaches the overall maximum of 2, as predicted in **Lemma** 2.4.*

We leave the optimality condition of the free (unconstrained) version to supplementary A.1.3.

## 3 SNAPHARD CONTRAST LEARNING

From the above analysis, it is evident that easy positives and negatives mainly act as fixation points that limit variability, whereas hard samples critically shape the optimization landscape. Motivated by this observation, we focus on the strategic generation and selection of hard positive and negative pairs, leading to our proposed **SnaPhArd** **C**ontrast **L**earning (*SPACL*) method.

**Generation and selection of hard positives ($\mathcal{P}_i^h$).** For the $i$-th original sample $\tilde{\mathbf{z}}_i$, we construct a candidate pool $\mathcal{C}_i = \{\tilde{\mathbf{z}}_i\} \cup \{\tilde{\mathbf{z}}_i^{(1)}, \ldots, \tilde{\mathbf{z}}_i^{(M)}\}$, consisting of $\tilde{\mathbf{z}}_i$ and its $M$ augmented variants. Prior methods typically adopt either (i) using $(\tilde{\mathbf{z}}_i, \tilde{\mathbf{z}}_i^{(m)})$ ($m \in [1, M]$) as an anchor-positive pair, or (ii) discarding $\tilde{\mathbf{z}}_i$ and pairing $(\tilde{\mathbf{z}}_i^{(m_1)}, \tilde{\mathbf{z}}_i^{(m_2)})$ ($m_1, m_2 \in [1, M]$). In contrast, we define the hardness of a candidate $\mathbf{u} \in \mathcal{C}_i$ by its average dissimilarity to the rest of the pool: $h(\mathbf{u}) := \frac{1}{|\mathcal{C}_i| - 1} \sum_{\mathbf{v} \in \mathcal{C}_i \setminus \{\mathbf{u}\}} d(\mathbf{u}, \mathbf{v})$, where $d(\cdot, \cdot)$ denotes a distance (or negative similarity) measure and is implemented as the inner product to align with the previous analysis (Eq. (2)). We then perform iterative selection to construct $\mathcal{P}_i^h$. Specifically, we first identify the anchor $\mathbf{z}_i = \arg\max_{\mathbf{u} \in \mathcal{C}_i} h(\mathbf{u})$, and $\mathcal{P}_i^h \leftarrow \{\mathbf{z}_i\}$. At each subsequent step, we choose $\mathbf{u}^* = \arg\max_{\mathbf{u} \in \mathcal{C}_i \setminus \mathcal{P}_i^h} \min_{\mathbf{v} \in \mathcal{P}_i^h} d(\mathbf{u}, \mathbf{v})$, and update $\mathcal{P}_i^h \leftarrow \mathcal{P}_i^h \cup \{\mathbf{u}^*\}$. This process repeats until $|\mathcal{P}_i^h| = \lambda_{\mathcal{P}^h}$, where $\lambda_{\mathcal{P}^h}$ ($< M$) is a hyperparameter controlling the number of selected hard positives.

**Remark 3.1.** *This farthest-point iterative selection maximizes the spread of $\mathcal{P}_i^h$ on the unit hypersphere, thereby enlarging $\mathrm{conv}(\mathcal{P}_i^h)$. In light of Theorem 2.1 and Corollary 2.2, this geometric*

*expansion mitigates collapse by preventing the anchor from coinciding with a single positive and works hand in hand with the choice of the negatives introduced next.*

**Generation and selection of hard negatives ($\mathcal{Q}_i^h$).** For the negatives of $\tilde{\mathbf{z}}_i$, standard strategies include (i) the in-batch scheme, where $\{\tilde{\mathbf{z}}_j\}_{j \neq i}$ are treated as negatives, or (ii) maintaining a momentum-based negative queue. Beyond these, we introduce an adversarial generator to further enrich the negative space. Specifically, the generator $G$ is encouraged to produce adversarial negatives $\tilde{\mathbf{z}}_i^-$ that closely resemble the hard positive set $\mathcal{P}_i^h$, while the discriminator $D$ is trained to distinguish between genuine positives and generated negatives. Formally, the adversarial min–max objective is given by

$$\min_G \max_D \; \mathbb{E}_{\mathbf{z}^+ \in \mathcal{P}_i^h}\big[\log D(\mathbf{z}^+)\big] + \mathbb{E}_{\tilde{\mathbf{z}}^- \sim G(\tilde{\mathbf{z}}_i)}\big[\log\big(1 - D(\tilde{\mathbf{z}}^-)\big)\big]. \tag{7}$$

The resulting candidate set of negatives is $\mathcal{Q}_i = \{\tilde{\mathbf{z}}_j \mid j \neq i\} \cup \{\tilde{\mathbf{z}}_i^-\}$. From $\mathcal{Q}_i$, we identify the hard negative subset $\mathcal{Q}_i^h \subset \mathcal{Q}_i$ characterized by *high* similarity to the anchor $\mathbf{z}_i$. Formally, for every $\mathbf{z}_i^{h,-} \in \mathcal{Q}_i^h, \forall \mathbf{z}_i^- \in \mathcal{Q}_i \setminus \mathcal{Q}_i^h$, we require

$$\begin{cases} \mathrm{sim}(f(\mathbf{z}_i), g(\mathbf{z}_i^{h,-})) > \mathrm{sim}(f(\mathbf{z}_i), g(\mathbf{z}_i^-)), \\ \displaystyle\sum_{\mathbf{z}_i^{h,-} \in \mathcal{Q}_i^h} \mathrm{sim}(f(\mathbf{z}_i), g(\mathbf{z}_i^{h,-})) = \lambda_{\mathcal{Q}^h} \sum_{\mathbf{z}_i^- \in \mathcal{Q}_i} \mathrm{sim}(f(\mathbf{z}_i), g(\mathbf{z}_i^-)), \end{cases} \tag{8}$$

where $\lambda_{\mathcal{Q}^h} \in (0, 1]$ is a screening coefficient controlling the *relative* proportion of hard negatives. It is worth emphasizing that while an **absolute** quantity is utilized for selecting hard positives, a **relative** threshold is adopted for hard negatives. This design choice is motivated by the fact that negatives are typically far more abundant and diverse than positives, and forcing a fixed number may either introduce many trivial negatives or overlook informative ones. Employing a relative threshold for negatives therefore ensures adaptability across diverse datasets while maintaining balance. The efficacy of this design is validated by ablation experiments, and the corresponding computational complexity is provided in Appendix A.4.

**Remark 3.2.** *The relative screening strategy for hard negatives $\mathcal{Q}_i^h$ effectively controls the contribution of negatives that lie closest to the positive convex hull $\mathrm{conv}(\mathcal{P}_i^h)$. From a geometric perspective, including too many easy negatives (those far from $\mathcal{P}_i^h$) increase the convex hull of the negatives and hence increase the chance of collapsing as shown in Theorem 2.1. In contrast, including only hard negatives ensures that the $\mathrm{conv}(\mathcal{Q}_i^h)$ lies close to $\mathrm{conv}(\mathcal{P}_i^h)$, thereby much likely to break the symmetric condition in (6), and hence aligns with Corollary 2.2, as the anchor cannot trivially project into the negative convex hull, which strengthens robustness against collapse and encourages more discriminative embeddings.*

**Overall loss.** Combining the above, the contrastive loss of *SPACL* is defined as:

$$\mathcal{L}_{SPACL} = -\sum_i \log \left\{ \frac{\sum_j^{|\mathcal{P}_i^h|} \exp\left(\langle f(\mathbf{z}_i), g(\mathbf{z}_i^{h,+})\rangle/\tau\right)}{\sum_j^{|\mathcal{P}_i^h|} \exp\left(\langle f(\mathbf{z}_i), g(\mathbf{z}_i^{h,+})\rangle/\tau\right) + \sum_j^{|\mathcal{Q}_i^h|} \exp\left(\langle f(\mathbf{z}_i), g(\mathbf{z}_i^{h,-})\rangle/\tau\right)} \right\}. \tag{9}$$

## 4 EXPERIMENTS

In this section, the proposed algorithm is evaluated on three downstream tasks across multi-modal scenarios, including image classification (*i.e.*, categorizing images into predefined classes), knowledge-graph link prediction (*i.e.*, inferring missing relations between entities) and out-of-domain intent detection(*i.e.*, determining whether an input matches known intents or is unseen). We utilize the inner product function to estimate the similarity. The screening terms for hard positive and negative are set as $\lambda_{\mathcal{P}^h} = 2$ and $\lambda_{\mathcal{Q}^h} = 0.95$, respectively (the impact from these two hyperparameters are discussed in the ablation study). All experiments are trained with a machine equipped with an NVIDIA A100 GPU server.

## 4.1 IMAGE CLASSIFICATION

**Datasets.** We evaluate our method on four widely used benchmarks: CIFAR-10, CIFAR-100, ImageNet-100, and ImageNet-1K. All methods, including baselines and *SPACL*, are implemented with the ResNet(-50) backbone to minimize the impact from encoder variations and focus primarily on the optimization loss[2]. In addition, we standardize the following settings: images are resized to $32 \times 32$ for CIFAR-10/100 and $224 \times 224$ for ImageNet-style datasets, and all models are trained for 200 epochs with a batch size of 256. Other hyperparameters follow the default configurations from original implementations of each baseline to ensure a fair comparison. Detailed dataset descriptions, baselines and their settings are provided in Appendix A.3.1. Their results are either directly resourced from original papers or reproduced using the officially released code.

Table 1: Comparison of baselines and *SPACL* across supervised, self-supervised, and weakly-supervised learning paradigms on CIFAR-10/100 and ImageNet-100/1K datasets.

| Category | Method | CIFAR10 | | CIFAR100 | | ImageNet-100 | | ImageNet-1K | |
|---|---|---|---|---|---|---|---|---|---|
| | | Top-1 | Top-5 | Top-1 | Top-5 | Top-1 | Top-5 | Top-1 | Top-5 |
| Supervised | Cross-Entropy | 95.07 | 99.82 | 74.01 | 91.89 | 83.17 | 95.78 | 78.20 | 93.71 |
| | SupCon | 95.51 | 99.85 | 76.57 | 93.50 | 85.06 | 96.84 | 78.72 | 94.31 |
| | VarCon | 95.94 | 99.87 | 78.29 | 93.59 | 86.34 | 96.96 | 79.36 | 94.37 |
| | *SPACL* (Ours) | **97.04**$_{\pm 0.04}$ | **99.92**$_{\pm 0.02}$ | **80.84**$_{\pm 0.06}$ | **94.32**$_{\pm 0.09}$ | **87.62**$_{\pm 0.09}$ | **97.84**$_{\pm 0.03}$ | **80.98**$_{\pm 0.12}$ | **95.45**$_{\pm 0.05}$ |
| Self-supervised | SimCLR | 91.52 | 99.78 | 70.67 | 92.01 | 71.54 | 91.56 | 70.31 | 90.37 |
| | MoCo V2 | 92.93 | 99.79 | 70.01 | 91.68 | 78.98 | 95.20 | 71.06 | 90.40 |
| | BYOL | 92.57 | 99.71 | 70.50 | 91.95 | 80.18 | 94.86 | 74.28 | 91.56 |
| | SwAV | 89.14 | 99.69 | 64.87 | 88.81 | 74.07 | 92.77 | 75.29 | 91.83 |
| | VicReg | 92.09 | 99.73 | 68.51 | 90.91 | 79.26 | 95.06 | 73.25 | 91.06 |
| | Barlow Twins | 92.70 | 99.80 | 71.02 | 91.95 | 80.83 | 95.24 | 73.26 | 91.10 |
| | *SPACL* (Ours) | **94.19**$_{\pm 0.07}$ | **99.87**$_{\pm 0.03}$ | **72.28**$_{\pm 0.05}$ | **94.39**$_{\pm 0.06}$ | **81.72**$_{\pm 0.04}$ | **95.55**$_{\pm 0.05}$ | **76.59**$_{\pm 0.09}$ | **93.41**$_{\pm 0.02}$ |
| Weakly-supervised | Grafit | 90.56 | 95.95 | 60.57 | 82.32 | 43.54 | 64.45 | 18.13 | 37.19 |
| | CoIns | 89.42 | 96.37 | 60.10 | 83.14 | 45.29 | 65.13 | 18.36 | 37.09 |
| | MaskCon | 91.63 | 98.72 | 65.52 | 83.64 | 47.74 | 66.98 | 19.08 | 38.17 |
| | *SPACL* (Ours) | **93.92**$_{\pm 0.04}$ | **99.51**$_{\pm 0.09}$ | **69.22**$_{\pm 0.09}$ | **86.00**$_{\pm 0.03}$ | **50.18**$_{\pm 0.11}$ | **70.51**$_{\pm 0.05}$ | **22.08**$_{\pm 0.05}$ | **40.17**$_{\pm 0.08}$ |

**Results.** Table 1 reports the classification accuracy on CIFAR-10, CIFAR-100, ImageNet-100, and ImageNet-1K under different supervision paradigms. Overall, our method achieves the best performance across all datasets and supervision settings. On average, *SPACL* improves Top-1 accuracy by 1.5 to 4.5% compared to the strongest baseline in each category, while also delivering consistent gains on Top-5 accuracy. For instance, under full supervision settings, *SPACL* reaches 97.04% Top-1 and 99.92% Top-5 on CIFAR-10, outperforming VarCon by +1.10% Top-1. On CIFAR-100, it attains 80.84% Top-1 and 94.32% Top-5, surpassing SupCon by +4.27% Top-1. On ImageNet-100, the method achieves 87.62% Top-1 and 97.84% Top-5, improving over VarCon by +1.28% Top-1. Finally, on full ImageNet-1K, *SPACL* delivers 80.98% Top-1 and 95.45% Top-5, which is +1.62% higher in Top-1 compared to VarCon, confirming its competitiveness even with large-scale benchmarks. These results confirm that *SPACL* provides an effective solution for visual representation learning under diverse learning paradigms.

## 4.2 LINK PREDICTION

**Datasets.** The proposed *SPACL* is evaluated on a multimodal Knowledge Graph (KG) completion task, *i.e.*, KG link prediction, where each entity is associated with both visual and textual modalities. We conduct experiments on three widely used benchmarks: WN9, FB15K-237, and FB15K. We follow standard evaluation metrics and report Mean Reciprocal Rank (MRR) together with Hits@1/3/10, which measure how highly the correct tail (or head) entity is ranked among all candidates. Followed prior work (Li et al., 2024b; Liu et al., 2024b), hyperparameters for *SPACL* are tuned by grid search. Specifically, the embedding dimension is searched over $\{100, 500, 1000, 1500, 2000\}$, the numbers of nearest neighbors is varied in $\{1, 2, 3, 6\}$, and the learning rate is drawn from $\{10^{-4}, 5 \times 10^{-4}, 10^{-3}, 10^{-2}, 10^{-1}, 2 \times 10^{-1}\}$. Detailed dataset descriptions, baselines and their implementation settings are deferred to Appendix A.3.2. Baseline results are directly resourced from original papers.

**Results.** As shown in Table 2, *SPACL* achieves the best prediction performance in terms of MRR and Hit@K. In particular, on FB15K-237 (where inverse relations are removed to eliminate trivial shortcuts, resulting in a sparser and more challenging dataset), the ability to identify informative hard samples becomes especially important. By explicitly targeting such hard positives and negatives, *SPACL* delivers consistent improvements, attaining an MRR of 41.30% and surpassing the strongest baseline LMKGE$_{l2}$ (40.00%) by +1.30%, while also raising Hit@10 to 61.20% compared to 57.80%

---

[2]An additional ablation study on different backbones is presented in Appendix A.5.

Table 2: Link prediction results on WN9, FB15K-237, and FB15K. We report MRR and Hit@K (K=1/3/10). A dash indicates the value is not available.

| Models | WN9 | | | | FB15K-237 | | | | FB15K | | | |
|--------|-----|------|------|-------|-----|------|------|-------|-----|------|------|-------|
| | MRR | Hit@1 | Hit@3 | Hit@10 | MRR | Hit@1 | Hit@3 | Hit@10 | MRR | Hit@1 | Hit@3 | Hit@10 |
| RESCAL | - | - | .708 | .447 | .356 | .263 | .393 | .541 | .644 | .544 | .708 | .824 |
| DistMult | .901 | .895 | .913 | .925 | .343 | .250 | .378 | .531 | .841 | .806 | .863 | .903 |
| TransE | .865 | .765 | .816 | .871 | .313 | .221 | .347 | .497 | .676 | .542 | .787 | .875 |
| RotatE | .910 | .901 | .915 | .926 | .333 | .240 | .368 | .522 | .783 | .727 | .820 | .877 |
| QuatE | - | - | - | - | .366 | .271 | .401 | .556 | .833 | .800 | .859 | .900 |
| ComplEx | .905 | .894 | .907 | .928 | .348 | .253 | .384 | .536 | .838 | .807 | .856 | .893 |
| IKRL | .901 | .904 | .912 | .928 | - | .194 | .284 | .458 | .594 | .484 | - | .768 |
| OTKGE | .923 | .911 | .930 | .947 | - | - | - | - | - | - | - | - |
| MKGformer | - | - | - | - | - | .256 | .367 | .504 | - | - | - | - |
| RSME | - | .878 | .912 | .923 | - | .242 | .344 | .467 | - | .802 | .881 | .924 |
| IMF | - | - | - | - | .389 | .287 | - | .593 | .837 | .785 | - | .914 |
| TransAE | .905 | .898 | .908 | .922 | - | .199 | .317 | .463 | - | - | - | .645 |
| MoSE | - | .909 | .937 | .967 | - | .281 | .411 | .565 | - | - | - | - |
| LMKGE | .934 | .917 | .944 | .969 | .400 | .306 | .437 | .585 | .868 | .840 | .885 | .918 |
| LMKGE$_{l2}$ | .931 | .914 | .940 | .967 | .394 | .302 | .431 | .578 | .869 | .842 | .885 | .918 |
| *SPACL* | .942$_{\pm 0.002}$ | .927$_{\pm 0.001}$ | .955$_{\pm 0.003}$ | .972$_{\pm 0.001}$ | .413$_{\pm 0.001}$ | .314$_{\pm 0.002}$ | .457$_{\pm 0.004}$ | .612$_{\pm 0.002}$ | .882$_{\pm 0.002}$ | .853$_{\pm 0.001}$ | .891$_{\pm 0.003}$ | .936$_{\pm 0.001}$ |

from LMKGE$_{l2}$. The result demonstrates that *SPACL* provides consistent gains across datasets of varying difficulty levels, validating its applicability to knowledge graph completion.

## 4.3 OUT-OF-DOMAIN INTENT DETECTION

**Dataset.** We also evaluate *SPACL* in a purely text-domain setting, where the goal is to detect out-of-domain (OOD) samples. Two widely used benchmarks, BANKING and StackOverflow, are employed. Following prior work (Zhou et al., 2023; Li et al., 2025), we randomly preserve 25%, 50%, and 75% of the total classes as in-domain (IND) classes, while the remaining classes serve as OOD. For *SPACL*, the Adam optimizer with a learning rate of $1e^{-5}$ is employed, and we conduct model training with a batch size of 8 over 20 epochs. The classification performance is evaluated using two metrics: F1-scores for OOD (F1-OOD) and IND (F1-IND) classes. Detailed dataset descriptions, baselines and their implementation settings are deferred to Appendix A.3.3. Baseline results are directly resourced from original papers or reproduced using the officially released code.

Table 3: OOD detection performances across numerous models on the BANKING and StackOverflow datasets, according to varying proportions (25%, 50%, 75%) of training classes.

| Ratio | Methods | BANKING (F1-OOD) | BANKING (F1-IND) | StackOverflow (F1-OOD) | StackOverflow (F1-IND) |
|-------|---------|------------------|------------------|------------------------|------------------------|
| 25% | MOGB | 88.29 | 74.50 | 94.42 | **87.80** |
| | TBOS | 92.70 | 79.70 | 95.52 | 85.35 |
| | *SPACL* | **93.51**±0.05 | **80.21**±0.09 | **96.72**±0.06 | 86.24±0.14 |
| 50% | MOGB | **89.71** | **87.27** | 81.04 | 81.53 |
| | TBOS | 84.42 | 84.63 | 90.41 | 88.96 |
| | *SPACL* | 85.48±0.03 | 85.57±0.18 | **91.52**±0.19 | **90.12**±0.08 |
| 75% | MOGB | 75.52 | 88.16 | 71.27 | 87.09 |
| | TBOS | 74.94 | 89.23 | 75.58 | 88.40 |
| | *SPACL* | **75.72**±0.02 | **89.95**±0.16 | **76.39**±0.18 | **88.96**±0.05 |

**Results.** As shown in Table 3, *SPACL* achieves competitive performance with recent baselines across the two datasets, achieving the best in-domain accuracy and OOD detection performance on average. For instance, for StackOverflow, *SPACL* achieves F1-OOD scores of 96.72 and 76.39 for 25% and 75% training classes, respectively. This presents a substantial improvement of 2.3 and 5.1 absolute points, respectively, over MOGB. These results clearly indicate that *SPACL* retains strong discriminative ability with purely textual data. Together with previous visual and multimodal experiments, it is worth nothing that the proposed *SPACL* algorithm is modality-agnostic and generalizes effectively across visual, textual, and multimodal CL scenarios.

## 4.4 ABLATION STUDY

We further conduct ablation studies to examine how different components of *SPACL* contribute to its performance. We also investigate the impact of the positive and negative sample budgets towards the final performance. Due to space constraints, these detailed experimental settings and results are provided in Appendix A.6.

**Model Breakdown.** To begin with, we analyze the individual components of *SPACL* to assess their relative contribution. Our method integrates four key aspects: (1) *anchor selection*, which maximizes the angular spread of positives by initializing with the hardest instance in the candidate pool;

(2) *hard positive selection*, which iteratively enlarges the convex hull of positives to enhance diversity and mitigate collapse; (3) *adversarial negatives*, where the adversarial training is adopted to produce hard negatives; and (4) *negative screening*, which applies a relative threshold to filter out trivial negatives. Accordingly, we construct the following variants for comparison: "w/o Anc" disables anchor selection via taking the original sample as the anchor; "w/o HP" removes farthest-point selection via taking random augmentation as positives; "w/o AN" discards adversarial negatives while "w/o NS" removes the screening mechanism.

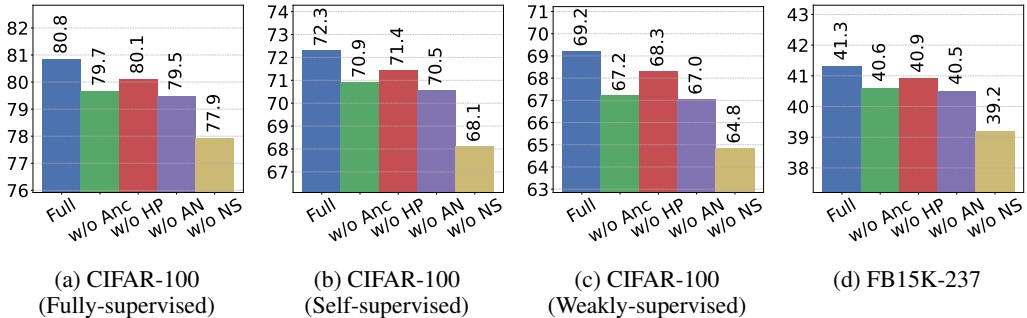

(a) CIFAR-100
(Fully-supervised)

(b) CIFAR-100
(Self-supervised)

(c) CIFAR-100
(Weakly-supervised)

(d) FB15K-237

Figure 1: Examination on individual components from *SPACL* on CIFAR-100 under fully-, self-, and weakly-supervised settings, and on FB15K-237.

Fig. 1 reports the results on CIFAR-100 and FB15K-237 across the four ablated variants, highlighting the contribution of each component. Overall, all variants underperform compared to the full model, confirming that every component of *SPACL* plays a positive role. The most significant performance drop occurs in "w/o NS", demonstrating the critical role of negative screening. That is, without screening, a large number of trivial negatives are retained, which expands the negative convex body and weakens the discriminative boundary. In addition, "w/o AN" and "w/o Anc" both lead to moderate decreases. The impact of "w/o Anc" reflects the role of anchor selection in maximizing angular spread, thereby reducing the risk of symmetry-induced collapse described in **Theorem** 2.1. Similarly, the drop in "w/o AN" shows that adversarially generated negatives help sharpen the boundary of the negative region, which is otherwise blurred when only in-batch or queued negatives are used. Finally, "w/o HP" results in the smallest decline, suggesting that farthest-point selection, while useful for enlarging the convex hull of positives and enhancing diversity, is less critical than maintaining a well-separated boundary between positives and negatives.

**Positive Pair Construction.** Next, we evaluate the impact of anchor and positive sample configurations. *SPACL* uses anchor selection and hard positive mining to improve diversity and avoid collapse. Accordingly, we construct two variants for comparison: "OAA" disables dynamic anchor selection by always choosing the original instance as the anchor and randomly sampling positives from its augmented views; and "AFP" draws both anchors and positives randomly from the augmented candidate pool.

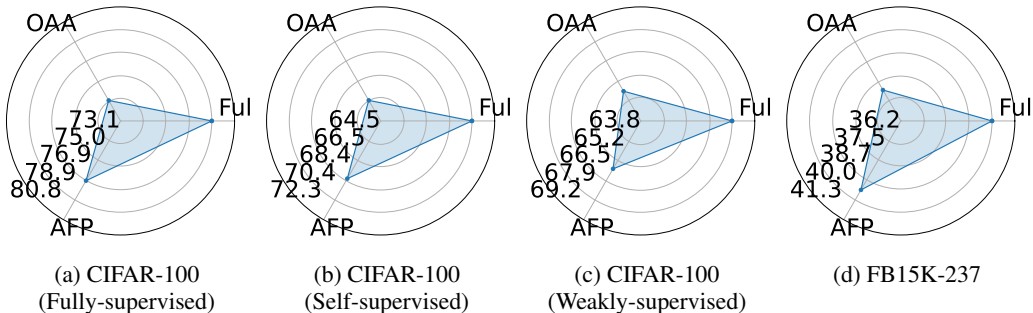

(a) CIFAR-100
(Fully-supervised)

(b) CIFAR-100
(Self-supervised)

(c) CIFAR-100
(Weakly-supervised)

(d) FB15K-237

Figure 2: Impact on the roles of dynamic anchor selection and hard positive mining, evaluated on CIFAR-100 under fully-, self-, and weakly-supervised paradigms, and on FB15K-237.

As shown in Fig. 2, both simplified variants perform worse than the full model, underscoring the importance of carefully designed positive pairs. The OAA variant shows a particularly large drop under self- and weakly-supervised settings, reflecting the limitation of using a fixed anchor. This observation is consistent with **Theorem** 2.1, which states that a static anchor reduces angular disper-

sion and compresses the positive convex hull. On the other hand, the AFP variant, while introducing anchor diversity, omits the structured expansion provided by farthest-point selection, resulting in less stable representation geometry. For instance, the experimental results show that on CIFAR-100 (self-supervised), measured by Top-1 accuracy, the full model achieves 72.28%, whereas OAA and AFP drop to 66.53% and 70.19%, respectively. On FB15K-237, using mean reciprocal rank (MRR) as the evaluation metric, the full model reaches 41.3%, while OAA drops to 38.2% and AFP reaches 40.7%. These results further confirm the role of both anchor dynamics and positive selection in maintaining discriminative geometry. Overall, these findings highlight the value of jointly optimizing anchors and positives, reinforcing the design motivation of *SPACL* 's positive construction.

**Adversarial Negatives vs In-Batch Negatives.** To assess the effect of negative sample quality, we compare two variants. The first, "IB," applies the contrastive loss directly to in-batch negatives, treating all co-occurring samples in the batch or queue as equally informative. The second, "IB+H," ranks the in-batch negatives and selects those with the highest similarity to the anchor, but does not adversarially generate new hard negatives. To isolate the effect of adversarial enhancement, both variants retain the same anchor selection and positive construction.

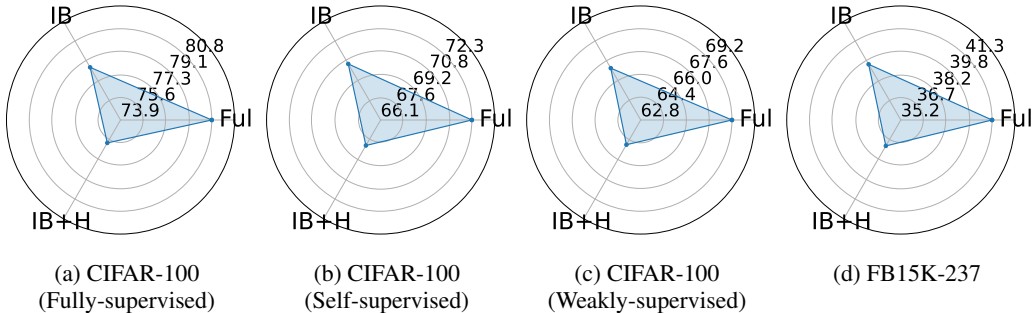

| (a) CIFAR-100 | (b) CIFAR-100 | (c) CIFAR-100 | (d) FB15K-237 |
| (Fully-supervised) | (Self-supervised) | (Weakly-supervised) | |

Figure 3: Comparison of adversarial negative strategies on CIFAR-100 under fully-, self-, and weakly-supervised settings, and on FB15K-237.

As shown in Fig. 3, the IB+H variant consistently achieves higher performance than the IB baseline across all benchmarks. This demonstrates that simply ranking and selecting the most informative in-batch negatives already improves contrastive learning by focusing on harder instances rather than treating all negatives equally. These results are also consistent with the "w/o AN" variant in the *Model Breakdown* analysis, confirming the role of negative quality in shaping the optimization landscape. The further gains of *SPACL* over IB+H highlight the additional benefit of adversarially generated boundary-aware negatives, showing that synthesized hard negatives complement original ones by tightening decision boundaries and strengthening discriminability.

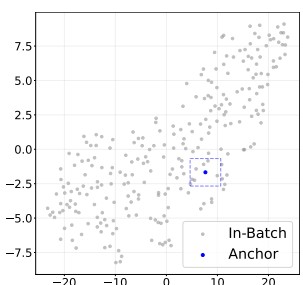 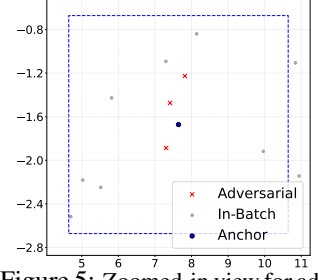 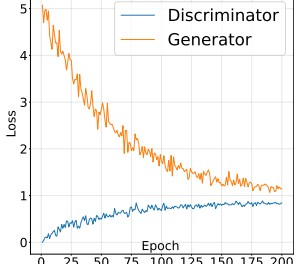

Figure 4: Embedding visualization of the anchor and in-batch negatives.

Figure 5: Zoomed-in view for adversarial and in-batch negatives of the same anchor.

Figure 6: Loss curve evolution of the generator and discriminator on CIFAR-100.

To further illustrate the effect of our adversarial negatives, we sample CIFAR-100 with a batch size of 256, randomly select one original sample as the anchor, generate three negatives using our adversarial module, and visualize all embeddings via t-SNE. As shown in Fig. 4 and 5, most in-batch negatives remain in distant clusters, whereas the adversarial negatives concentrate in the region surrounding the anchor, indicating that adversarial negatives effectively tighten the local decision boundary and potentially enhance the model capability.

We also observe that our proposed adversarial module, to some extent, mitigates the impact of false negatives. False negatives typically refer to samples that are semantically related to the anchor (with high-similarity) but are mistakenly treated as negatives. Importantly, our adversarially generated negatives help construct stronger and more informative hard negatives than those obtained via in-batch sampling, including potential false negatives, as demonstrated in Fig. 4 and 5. As these adversarial negatives exhibit higher similarity, the proposed selection mechanism naturally prioritizes them, thereby reducing the likelihood that moderately similar samples (*i.e.*, some potential false negatives) are selected as hard negatives. Yet, we note that explicitly identifying or eliminating false negatives is beyond the scope of this work, and we leave this as future work.

In addition, to verify the stability of adversarial training, we also visualize both the generator and discriminator losses on the CIFAR-100 task, as shown in Fig. 6. The discriminator loss exhibits a smooth, gradually increasing trend, while the generator loss decreases throughout training. We attribute this result to the following reason: early in training, the generator might produce weaker adversarial examples, making them easy for the discriminator to classify, thereby resulting in high generator loss and low discriminator loss. As training progresses, the generator becomes stronger and produces harder samples, causing the discriminator loss to increase, while the generator loss continues to decrease as it adapts to the discriminator. Importantly, both components exhibit stability and are optimized smoothly.

## 5 CONCLUSION

Contrastive Learning (CL) enforces proximity between anchors and positives while separating negatives, yet the theoretical basis of pair sampling and its impact on performance remain underexplored. This paper introduces the SnaPhArd Contrast Learning (*SPACL*) algorithm, which strategically screens hard contrastive pairs. Our method highlights the role of hard samples during constructing contrastive pairs and computing the contrastive loss. Thorough theoretical analysis justifies hard sampling and discusses optimality conditions and potential collapsing. Empirical evaluation demonstrates the effectiveness of *SPACL*. Future research could explore different strategies for measuring sample difficulty and exploring alternative perturbation methods. Additionally, extending our analysis to negative-free CL could further illustrate its adaptability and broader its applicability.

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

# A SUPPLEMENTARY MATERIALS

## A.1 DETAILED ANALYSIS

### A.1.1 PROOF FOR THEOREM 2.1

*Proof.* The Lagrangian $\mathcal{L}_c$ from Eq. (4) is

$$\mathcal{L}_c = \frac{\sum_j \exp\langle \mathbf{x}, \mathbf{n}_j \rangle}{\sum_j \exp\langle \mathbf{x}, \mathbf{p}_j \rangle} - \lambda(\|\mathbf{x}\|^2 - 1),$$

where $\lambda \in \mathbb{R}$ is the Lagrangian multiplier. The optimality conditions can be defined as Eq. (10).

$$\begin{cases} \frac{(\sum_j \exp\langle \mathbf{x}, \mathbf{p}_j \rangle)(\sum_j \exp\langle \mathbf{x}, \mathbf{n}_j \rangle \mathbf{n}_j) - (\sum_j \exp\langle \mathbf{x}, \mathbf{n}_j \rangle)(\sum_j \exp\langle \mathbf{x}, \mathbf{p}_j \rangle \mathbf{p}_j)}{(\sum_j \exp\langle \mathbf{x}, \mathbf{p}_j \rangle)^2} - 2\lambda\mathbf{x} = 0 \\ \|\mathbf{x}\|^2 = 1 \end{cases}. \quad (10)$$

Note that here $\mathbf{x}$ is meant to be the optimal solution, conventionally written as $\mathbf{x}^*$. The second equation signifies the spherical distribution condition, *i.e.*, $\mathbf{x} \in \mathcal{S}^{m-1}$, where $m$ denotes the dimension of the representation space, *i.e.*, $\mathbb{R}^m$, and $\mathcal{S}^{m-1}$ is the sphere embedded in $\mathbb{R}^m$ with the dimension of $m - 1$. We can further derive

$$\sum_j \frac{\exp\langle \mathbf{x}, \mathbf{n}_j \rangle}{s^-} \mathbf{n}_j - \sum_j \frac{\exp\langle \mathbf{x}, \mathbf{p}_j \rangle}{s^+} \mathbf{p}_j = \frac{2\lambda s^+}{s^-} \mathbf{x}, \quad (11)$$

due to the non-negativity property of the sum of exponentials, *i.e.*, $s^-$ and $s^+$ are strictly positive. Note that $\sum_j \exp\langle \mathbf{x}, \mathbf{n}_j \rangle / s^- = 1$, and $\sum_j \exp\langle \mathbf{x}, \mathbf{p}_j \rangle / s^+ = 1$. It indicates the L.H.S of Eq. (11) represents two *convex* reconstructions. To simplify, we introduce the definition of a convex reconstruction for a set of arbitrary vectors $\mathbf{v}_i$'s ($N$ is the set size):

$$conv\{\mathbf{v}_i\}_{i=1}^N = \{\mathbf{a} | \exists w_i \in \mathbb{R}, i = 1, \ldots, N, \text{s.t. } \mathbf{a} = \sum_i w_i \mathbf{v}_i, \sum_i w_i = 1, \text{ and } w_i \geq 0\ \forall i\}. \quad (12)$$

Then, we have:

$$\begin{cases} \mathbf{n}(\mathbf{x}) = \sum_j \frac{\exp\langle \mathbf{x}, \mathbf{n}_j \rangle}{s^-} \mathbf{n}_j, & \mathbf{n}(\mathbf{x}) \in conv\{\mathbf{n}_j\}_{j=1}^{N^-}, \\ \mathbf{p}(\mathbf{x}) = \sum_j \frac{\exp\langle \mathbf{x}, \mathbf{p}_j \rangle}{s^+} \mathbf{p}_j, & \mathbf{p}(\mathbf{x}) \in conv\{\mathbf{p}_j\}_{j=1}^{N^+}, \end{cases}$$

where $\mathbf{n}(\mathbf{x})$ and $\mathbf{p}(\mathbf{x})$ are functions dependent on $\mathbf{x}$. The optimality condition from Eq. (11) can then be simplified to:

$$\mathbf{n}(\mathbf{x}) - \mathbf{p}(\mathbf{x}) = c\mathbf{x}, \quad (13)$$

where $c = (2\lambda s^+)/s^-$ ensures equality by choosing an appropriate value for $\lambda$.

When there is only a single positive, $\mathbf{p}(x) = \mathbf{p}$. From Eq. (13), it is easy to see that if $\mathbf{x} = \mathbf{p}$, then $\mathbf{n}(\mathbf{x}) = (1 + c)\mathbf{p}$. Hence, by setting $C = 1 + 2\lambda/s^-$, and $s^+ = 1$, we establish the necessary condition. The argument holds in both directions, confirming the "if and only if" scenario. $\square$

Fig. 7 further illustrates the optimality condition for the case of multiple positives. Note that both $\mathbf{p}(\mathbf{x})$ and $\mathbf{n}(\mathbf{x})$ must exist within the convex hulls of positives and negatives, respectively, despite the position of the optimal solution $\mathbf{x}$ in $\mathcal{S}^{m-1}$.

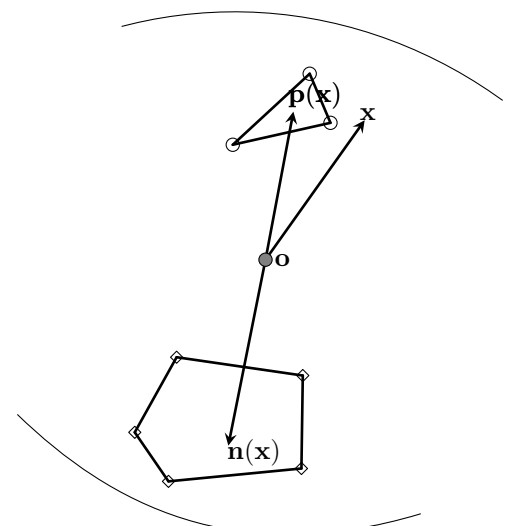

Figure 7: Illustration of optimality condition. The polygons tightly around positives (annotated as circles) and negatives (annotated as diamonds) are $conv\{\mathbf{p}_i\}_{i=1}^{N^+}$ and $conv\{\mathbf{n}_i\}_{i=1}^{N^-}$ respectively.

### A.1.2 PROOF FOR LEMMA 2.4

*Proof.* We follow Eq. (3) and expand the part before $\diamondsuit$, recognizing it is precisely the linear operator $\nabla_{\mathbf{x}}\mathcal{L}_c$. Then we have the following computation by omitting the index $i$:

$$
\begin{aligned}
\nabla_{\mathbf{x}}\mathcal{L}_c &= \frac{1}{s^-/s^+ + 1}\frac{\partial}{\partial\mathbf{x}}\left(\frac{s^-}{s^+}\right)\\
&= \frac{s^-}{s^+ + s^-}\left(\sum_j\frac{\exp\langle\mathbf{x},\mathbf{n}_j\rangle}{s^-}\mathbf{n}_j - \sum_j\frac{\exp\langle\mathbf{x},\mathbf{p}_j\rangle}{s^+}\mathbf{p}_j\right)\\
&= \frac{1}{\frac{s^+}{s^-}+1}\left(conv\{\mathbf{n}_j\}_{j=1}^{N^-} - conv\{\mathbf{p}_j\}_{j=1}^{N^+}\right)\\
&= \frac{1}{\frac{s^+}{s^-}+1}\mathbf{v}\quad(\text{let }\mathbf{v}=conv\{\mathbf{n}_j\}_{j=1}^{N^-} - conv\{\mathbf{p}_j\}_{j=1}^{N^+}).
\end{aligned}
\tag{14}
$$

Subsequently, $\|\nabla_{\mathbf{x}}\mathcal{L}_c\|$ is bounded by:

$$
\|\nabla_{\mathbf{x}}\mathcal{L}_c\| \leq \frac{2}{\frac{c}{N^-e}+1},
\tag{15}
$$

where we apply the facts that $\|\mathbf{v}\| \leq 2$, $s^+ \leq N^+e^{-1} = c$ for a fixed $N^+$, and all vectors are constrained to be on $\mathcal{S}^{m-1}$, ensuring $\max s^- = N^-e$. The claimed limit follows. $\qquad\square$

Fig. 8 illustrates the length bound as a function of $N^+$ values. Notably, when $N^+ > 1$, the initial value of $\|\nabla_{\mathbf{x}}\mathcal{L}_c\|$ experiences a slight reduction, followed by a more substantial increase as $N^-$ increases, and the convergence for the gradient length is rapid. In particular, when $N^- = 1000$, $\|\nabla_{\mathbf{x}}\mathcal{L}_c\| \approx 2$ applied for all cases (as easily derived from Eq. (15)). This offers another rationale for hard sample selection: ensure the most effective minimum $s^+/s^-$ in Eq. (14), to achieve the maximum gradient norm without incurring unnecessary computational overhead.

### A.1.3 OPTIMALITY CONDITION FOR FREE (UNCONSTRAINED) VERSION

Now we turn to the alternative objective without constraints, *i.e.*, the free version:

$$
\mathcal{L}_f = \frac{\sum_j\exp\left\langle\frac{\tilde{\mathbf{x}}}{\|\tilde{\mathbf{x}}\|},\mathbf{n}_j\right\rangle}{\sum_j\exp\left\langle\frac{\tilde{\mathbf{x}}}{\|\tilde{\mathbf{x}}\|},\mathbf{p}_j\right\rangle} \equiv \frac{\sum_j\exp\langle\mathbf{x},\mathbf{n}_j\rangle}{\sum_j\exp\langle\mathbf{x},\mathbf{p}_j\rangle} = \frac{s^-}{s^+},
\tag{16}
$$

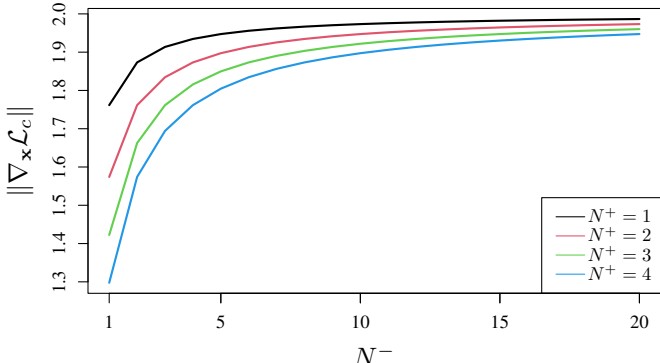

Figure 8: Maximum $\|\nabla_{\mathbf{x}}\mathcal{L}\|$ as a function of $N^-$ for various $N^+$ values.

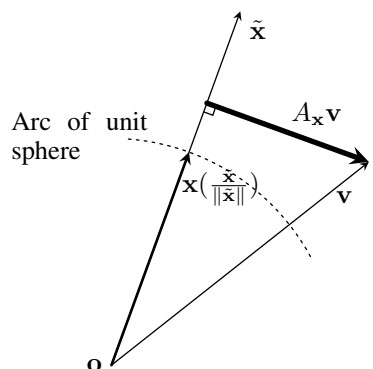

Figure 9: Illustration of the orthogonal projection $A_{\mathbf{x}}$.

where $\mathbf{x} = \tilde{\mathbf{x}}/\|\tilde{\mathbf{x}}\|$ and $\|\mathbf{x}\| = 1$. The difference to the constrained objective lies:

$$
\begin{aligned}
\frac{\partial \mathcal{L}_f}{\partial \tilde{\mathbf{x}}} &= \frac{s^+ \sum_j \exp\left\langle \frac{\tilde{\mathbf{x}}}{\|\tilde{\mathbf{x}}\|}, \mathbf{n}_j \right\rangle \left(\frac{\mathbf{I}}{\|\tilde{\mathbf{x}}\|} - \frac{\tilde{\mathbf{x}}\tilde{\mathbf{x}}^\top}{\|\tilde{\mathbf{x}}\|^3}\right)\mathbf{n}_j}{(s^+)^2} \\
&\quad - \frac{s^- \sum_j \exp\left\langle \frac{\tilde{\mathbf{x}}}{\|\tilde{\mathbf{x}}\|}, \mathbf{p}_j \right\rangle \left(\frac{\mathbf{I}}{\|\tilde{\mathbf{x}}\|} - \frac{\tilde{\mathbf{x}}\tilde{\mathbf{x}}^\top}{\|\tilde{\mathbf{x}}\|^3}\right)\mathbf{p}_j}{(s^+)^2} \\
&= \frac{s^+ \sum_j \exp\langle \mathbf{x}, \mathbf{n}_j \rangle (\mathbf{I} - \mathbf{x}\mathbf{x}^\top)\mathbf{n}_j}{\|\tilde{\mathbf{x}}\|(s^+)^2} \\
&\quad - \frac{s^- \sum_j \exp\langle \mathbf{x}, \mathbf{p}_j \rangle (\mathbf{I} - \mathbf{x}\mathbf{x}^\top)\mathbf{p}_j}{\|\tilde{\mathbf{x}}\|(s^+)^2},
\end{aligned}
$$

with the application of the following identity

$$
\frac{\partial \frac{\mathbf{x}}{\|\mathbf{x}\|}}{\partial \mathbf{x}} = \frac{\mathbf{I}}{\|\mathbf{x}\|} - \frac{\mathbf{x}\mathbf{x}^\top}{\|\mathbf{x}\|^3},
$$

where $\mathbf{I}$ is the identity matrix with compatible dimensions.

Write $A_{\mathbf{x}} := \mathbf{I} - \tilde{\mathbf{x}}\tilde{\mathbf{x}}^\top/\|\tilde{\mathbf{x}}\|^2 = \mathbf{I} - \mathbf{x}\mathbf{x}^\top$ and $w_j^{+/-}(\mathbf{x}) = \exp\langle \mathbf{x}/\|\mathbf{x}\|, \mathbf{p}_j/\mathbf{n}_j \rangle /s^{+/-} = \exp\langle \mathbf{x}, \mathbf{p}_j/\mathbf{n}_j \rangle /s^{+/-}$. The optimality condition gives $\sum_j w_j^-(\mathbf{x})A_{\mathbf{x}}\mathbf{n}_j = \sum_j w_j^+(\mathbf{x})A_{\mathbf{x}}\mathbf{p}_j$. Apparently $A_{\mathbf{x}}$ is a linear map that converts a vector to the one perpendicular to $\mathbf{x}$. That is, $A_{\mathbf{x}}\mathbf{v} = \mathbf{v} - \mathbf{x}^\top\mathbf{v}\mathbf{x} = \mathbf{v} - (\tilde{\mathbf{x}}/\|\tilde{\mathbf{x}}\|)^\top \mathbf{v} \cdot \tilde{\mathbf{x}}/\|\tilde{\mathbf{x}}\|$ as illustrated in Fig. 9. Therefore, $A_{\mathbf{x}}\mathbf{v}$ resides in the hyperplane centered at the origin $\mathbf{o}$, denoted as $P_{\mathbf{x}}$. This hyperplane has a normal vector $\mathbf{x}$ and an apparent dimensionality of $m - 1$. Consolidating these observations, we have the following straightforward optimality condition:

$$
\tilde{\mathbf{n}}(\tilde{\mathbf{x}}) = \tilde{\mathbf{p}}(\tilde{\mathbf{x}}), \tag{17}
$$

where $\tilde{\mathbf{n}}(\tilde{\mathbf{x}}) = \sum w_j^-(\mathbf{x})\tilde{\mathbf{n}}_j$, $\tilde{\mathbf{n}}_j = A_\mathbf{x}\mathbf{n}_j$, and likewise for $\tilde{\mathbf{p}}(\tilde{\mathbf{x}})$. We can see that $\tilde{\mathbf{n}}(\tilde{\mathbf{x}}) \in conv\{\tilde{\mathbf{n}}_i\}_{i=1}^{N^-}$ and $\tilde{\mathbf{p}}(\tilde{\mathbf{x}}) \in conv\{\tilde{\mathbf{p}}_i\}_{i=1}^{N^+}$, analogous to Eq. (13). The difference is the convex hulls are now in $P_\mathbf{x}$ orthogonal to $\mathbf{x}$, instead of in ambient space $\mathbb{R}^m$.

The optimality condition in Eq. (17) appears distinct from Eq. (13). However, they are closely related. We can express Eq. (17) to separate $\tilde{\mathbf{x}}$ or $\mathbf{x}$, *i.e.*,

$$
\begin{aligned}
\tilde{\mathbf{n}}(\tilde{\mathbf{x}}) &= \sum_j \frac{\exp\langle \mathbf{x}, \mathbf{n}_j \rangle}{s^-}\mathbf{n}_j - \sum_j \frac{\exp\langle \mathbf{x}, \mathbf{n}_j \rangle \langle \mathbf{x}, \mathbf{n}_j \rangle}{s^-}\mathbf{x} \\
&= \mathbf{n}(\mathbf{x}) - \sum_j \frac{\exp\langle \mathbf{x}, \mathbf{n}_j \rangle \langle \mathbf{x}, \mathbf{n}_j \rangle}{s^-}\mathbf{x} \\
&= \mathbf{n}(\mathbf{x}) - \tilde{s}^-\mathbf{x} \ (\text{write } \tilde{s}^- = \sum_j \frac{\exp\langle \mathbf{x}, \mathbf{n}_j \rangle \langle \mathbf{x}, \mathbf{n}_j \rangle}{s^-}).
\end{aligned}
$$

Similarly one can have $\tilde{\mathbf{p}}(\tilde{\mathbf{x}}) = \mathbf{p}(\mathbf{x}) - \tilde{s}^+\mathbf{x}$. The optimality condition derived from Eq. (17) can be reformulated as $\mathbf{n}(\mathbf{x}) - \mathbf{p}(\mathbf{x}) = (\tilde{s}^+ - \tilde{s}^-)\mathbf{x} = c\mathbf{x}$, which is nearly identical to Eq. (13), except for the term $c$. Nonetheless, the constrained version has a straightforward geometric interpretation that can be easily visualized, as demonstrated in Fig. 7, while the free version involves an orthogonal projection onto $P_\mathbf{x}$.

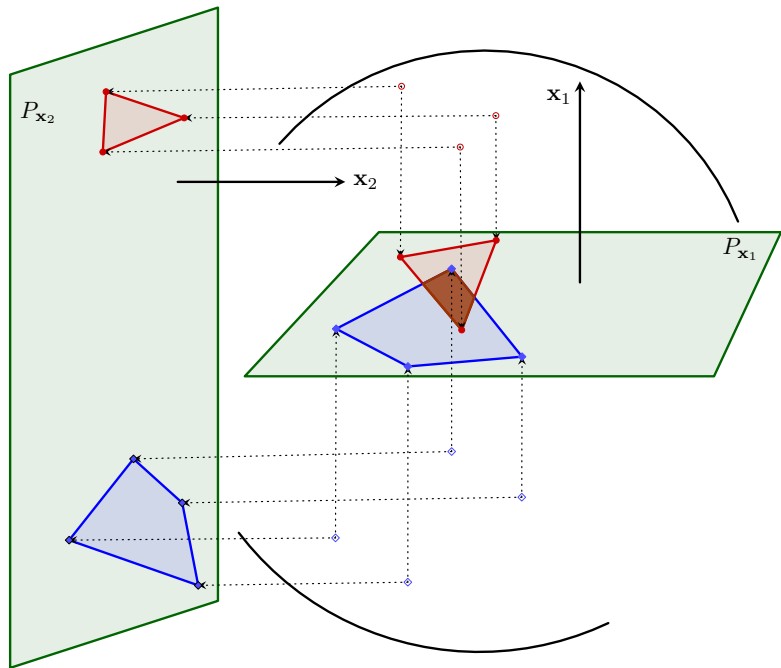

Figure 10: Illustration of optimality condition from the free version for $\mathbf{x}_1$ and $\mathbf{x}_2$. Hollow circles are positives and hollow diamonds are negatives. Corresponding solid ones are projections to $P_{\mathbf{x}_1}$ and $P_{\mathbf{x}_2}$. The polygons are convex hulls of projected positives and negatives. Dotted lines show the projection direction. The optimal solution has to lie within the intersection of the convex bodies.

Fig. 10 illustrates the optimality conditions for $\mathbf{x}_1$ and $\mathbf{x}_2$ (two candidates of $\mathbf{x}$). Apparently $\mathbf{x}_2$ cannot be the solution because the convex bodies of negatives and positives in $P_{\mathbf{x}_2}$ has no intersection. Although $\mathbf{x}_1$ is a potential candidate for the solution since convex hulls intersect in $P_{\mathbf{x}_1}$, the condition in Eq. (17) must also be satisfied.

## A.2 RELATED WORK

Contrastive Learning (CL) has garnered significant attention for leveraging input data as an additional supervisory signal during model training (Chen et al., 2020a; He et al., 2020; Grill et al., 2020; Chen & He, 2021). Specifically, given an anchor sample $z$ and an encoder function $f(\cdot)$, the

general form of the contrastive loss is defined as follows:

$$\text{sim}\left(f(\mathbf{z}), f(\mathbf{z}^+)\right) \gg \text{sim}\left(f(\mathbf{z}), f(\mathbf{z}^-)\right), \tag{18}$$

where $\text{sim}(\cdot, \cdot)$ represents a user-defined similarity function, and $\mathbf{z}^+$ and $\mathbf{z}^-$ represent *contrastive pairs* of positive and negative instances *w.r.t* $\mathbf{z}$, respectively. The effectiveness of CL hinges on the ability to ensure positive samples remain closely to anchors in the latent space, while negative ones are pushed apart. Recent years have witnessed remarkable progress in CL, involving a wide range of techniques for the **generation and sampling of positive/negative instances**.

### A.2.1 GENERATION OF CONTRASTIVE PAIRS

Contrastive pairs, also referred to as positive and negative samples, are of paramount importance in CL and serve as the foundation for the model training. Extensive research into their generation, including minibatch adaptation, data augmentation, and knowledge-based methods, has significantly improved the data quality and diversity.

**Minibatch adaptation** primarily caches current or past in-batch data as positive/negative samples. The MoCo (Momentum Contrast) series, exemplified by **MoCo-V1** (He et al., 2020) and **MoCo-V2** (Chen et al., 2020b), employs a momentum-based queue to accumulate negatives from past minibatches. In contrast, **MoCo-V3** (Chen et al., 2021) and **SimCLR** (Chen et al., 2020a) abandon this separate queue, instead using augmented versions of different samples (not anchors) from the current minibatch as negatives.

**Data augmentation** involves instance- and feature-level transformations applied to original training samples. For *instance-level* augmentation in Computer Vision (Kupyn & Rupprecht, 2025; Wang et al., 2024b; Feng et al., 2024), common techniques include In Natural Language Processing (Peng et al., 2024; Lee et al., 2024a; Evuru et al., 2024), augmentation techniques might involve translation, rephrasing, synonym replacement, and *etc*. Some studies consider adversarial samples (Zhang et al., 2025; Liu et al., 2023), via dynamically generating negative samples as a min-max problem. For *feature-level* augmentation, **HNCSE** (Liu et al., 2025b) performs feature-level augmentation by mixing hard negatives in the embedding space and applying positive mixing guided by the hardest negative, further enriching contrastive pair diversity and difficulty for sentence embeddings. Moreover, **STRFE** (Bao et al., 2024) co-augments structure and node features in the spectral domain—boosting low-frequency components for structural consistency while enhancing high-frequency components for feature-level diversity—to yield harder negatives and stronger representations. Complementarily, **RA-CC** (Sun et al., 2025) combines linear and model-based nonlinear feature-level augmentations and mines visual confusion categories to reweight negatives, thereby producing harder views and sharpening cluster discrimination. **MoCo-FT** (Zhu et al., 2021) proposes feature extrapolation and interpolation (a linear combination) of positives and negatives to create diverse contrastive pairs. Similarly, the **MixCSE** model (Zhang et al., 2022d) mixes positive and negative samples during the training process. **M-Mix** (Zhang et al., 2022c) employs Graph Neural Networks to adaptively select multiple neighboring nodes, assigning them dynamic weights for generating negatives. Another line of research discards negatives and relies exclusively on positives. Two notable approaches, **BYOL** (Grill et al., 2020) and **SimSiam** (Chen & He, 2021), employ a stop-gradient operation to construct positives as a low-pass-filtered version of anchor samples. A specific study by Zhang *et al.* (Zhang et al., 2022a) later provides a comprehensive analysis of SimSiam. Two key factors, known as *de-centering* and *dimensional de-correlation*, are identified to prevent solution collapsing within the context of negative-free CL.

In contrast to the aforementioned minibatch adaptation and data augmentation, **knowledge-based methods** leverage external knowledge and prior information. For example, Ji *et al.* (Ji et al., 2023) propose utilizing the social networking relationships among users and the commonality relationships of items to generate positive samples for online recommendation. In machine reading comprehension, cloze-like question-answer pairs are generated as positives (Liu et al., 2022; Hu et al., 2022). Furthermore, prior knowledge about co-occurring events is utilized in (Gao et al., 2022) to form contrastive pairs.

### A.2.2 SAMPLING OF CONTRASTIVE PAIRS

In contrast to the generation of contrastive pairs, the sampling technique aims to select **high-quality** or **hard** positive and negative samples. In the context of CL, sample hardness can be defined by the

feature proximity of a negative sample to that of a positive sample, and vice versa. In other words, the closeness in the representation space between contrastive pairs is a key criterion for determining sample hardness. The more similar a contrastive pair is, the harder it is for a model to differentiate them, which, in turn, enforces the model to learn/produce more discriminative representations.

To begin with, *adaptive sampling* is leveraged in graph CL to reweight negatives toward the most informative ones while suppressing false negatives via polarization regularization (Wan et al., 2024). In a complementary vein, a *curricular weighting* strategy gradually shifts attention from easy to hard negatives to improve convergence and robustness to false negatives (Zhuang et al., 2024b). From a theoretical angle, hard-negative mining is further justified through an isometric-approximation view that clarifies when mining avoids collapse and why it benefits metric learning (Xu et al., 2022). Likewise, *difficulty-based sampling* debiases contrastive learning by emphasizing bias-conflicting hard examples using a triplet-driven biased encoder (Jang & Wang, 2023). Extending these ideas to software engineering, *uncertainty-aware embeddings* are coupled with *hard negative sampling* to construct challenging query–code pairs and strengthen inter-/intra-modality alignment (Liu et al., 2025a). Meanwhile, in contrastive graph clustering, *enhanced hard sample mining with cluster-guiding* explicitly repels hard negatives while attracting hard positives via pseudo-labels (Li et al., 2024a). Finally, a *multi-scale* hard sample mining mechanism upweights difficult pairs across granularities to reinforce clustering performance (Ren et al., 2025).

## A.3 EXPERIMENTAL DETAILS

We provide experimental setups on three benchmarks to validate the generality of our method. These tasks span vision, textual, and multimodal data, covering low- to high-resolution image recognition, textual understanding, and multimodal relational reasoning. Together, they demonstrate that the proposed *SPACL* algorithm consistently adapts across diverse modalities, dataset scales, and supervision regimes.

### A.3.1 IMAGE CLASSIFICATION: DATASETS AND IMPLEMENTATION DETAILS

**Datasets.** We evaluate the proposed method on four widely used benchmark datasets: **CIFAR-10**, **CIFAR-100**, **ImageNet-100**, and **ImageNet-1K**. CIFAR-10 and CIFAR-100 consist of 60,000 color images of size $32 \times 32$, with CIFAR-10 containing 10 classes (6,000 images per class) and CIFAR-100 containing 100 classes (600 images per class). ImageNet-100 is a medium-scale subset of ImageNet, covering 100 categories with 1,300 training images and 50 validation images per class. ImageNet-1K includes more than 1.2 million training images and 50,000 validation images spanning 1,000 categories. Together, these datasets span a wide range of scales from low-resolution small-scale tasks (CIFAR-10/100) to large-scale high-resolution recognition (ImageNet-100/1K), providing a comprehensive basis for evaluating classification performance in contrastive learning.

**Implementation.** To demonstrate the generality of our framework, we conduct experiments under three supervision settings: fully-supervised, self-supervised, and weakly-supervised learning. This design enables a thorough assessment of the robustness and adaptability of our method across diverse training scenarios.

For fully-supervised settings, we consider **Cross-Entropy**, the conventional training paradigm using classification loss, **SupCon** (Khosla et al., 2020), which enhances representation learning by enforcing intra-class similarity and inter-class separation, and **VarCon**(Wang et al., 2025), a recent extension that introduces variational modeling into contrastive learning.

In the self-supervised category, we evaluate several state-of-the-art methods. **SimCLR** (Chen et al., 2020a) and **MoCo V2** (Chen et al., 2020b) employ instance discrimination with different design choices for negative sampling and momentum encoders. **BYOL** (Grill et al., 2020) removes explicit negatives and learns by aligning predictions with target networks. **SwAV** (Caron et al., 2020) combines contrastive objectives with online clustering to assign prototypes. **VicReg** (Bardes et al., 2022) regularizes variance, invariance, and covariance to improve feature robustness. **Barlow Twins** (Zbontar et al., 2021) minimizes redundancy between representation dimensions to enforce richer feature decorrelation.

For weakly-supervised baselines, we include methods that integrate limited label information with contrastive signals. **Grafit** (Touvron et al., 2021) couples supervised contrastive learning with instance-level self-supervision to mitigate under-clustering. **CoIns** (Xu et al., 2021) combines

cross-entropy with contrastive objectives to balance label guidance and representation diversity. **MaskCon** (Feng & Patras, 2023) leverages coarse labels through a masked contrastive mechanism, refining inter-sample relations by filtering out noisy associations.

### A.3.2 LINK PREDICTION: DATASETS AND IMPLEMENTATION DETAILS

**Datasets.** Evaluations are conducted on three widely used benchmarks, **FB15K**, **FB15K-237**, and **WN9**. *FB15K* originates from Freebase and provides a medium-scale relational graph that is often augmented with multimodal side information (entity-linked images and short textual descriptions). In our setting, it contains 14,951 entities, 1,345 relation types, and 592,213 triples, split into 483,142 training, 50,000 validation, and 59,071 test triples. *FB15K-237* is derived from FB15K by removing inverse and redundant relations, yielding a harder and more reliable benchmark for generalization while largely preserving the entity universe and mitigating relation leakage, it consists contains 14,541 entities, 237 relation types, and 272,115 / 17,535 / 20,466 triples for training, validation, and test, respectively. *WN9* is a small Knowledge Graph extracted from WordNet with 9 relation types and 6,555 entities, yielding 11,741 / 1,337 / 1,319 triples for training, validation, and test.

**Implementation.** We adopt the standard Knowledge Graph completion setting. That is, for each test triple $(h, r, t)$, our inputs or queries can be $(h, r, ?)$ or $(?, r, t)$, where "?" denotes the unknown entity. We then use the trained model to score all candidate entities and rank them accordingly. Additionally, the Mean Reciprocal Rank (MRR) and Hit Ratio at cutoffs 1, 3, and 10 (Hit@1/3/10) are employed as our evaluation metrics. MRR measures the average reciprocal rank of the correct entity in the prediction list, thus emphasizing how early the correct answer appears. Hit@k computes the proportion of test queries for which the correct entity is ranked within the top $k$ positions, with Hit@1, Hit@3, and Hit@10 reflecting increasingly relaxed success criteria. In all cases, larger values indicate better link prediction performance.

We compare *SPACL* with a broad set of baselines covering both unimodal and multimodal Knowledge Graph embedding approaches. The unimodal group consists of tensor factorization models such as RESCAL (Nickel et al., 2011), bilinear formulations like DistMult (Yang et al., 2015) and ComplEx (Trouillon et al., 2016), and translational distance models including TransE (Bordes et al., 2013), RotatE (Sun et al., 2019), and QuatE (ZHANG et al., 2019). These methods capture relational structure solely from symbolic triplets. The multimodal group augments structural embeddings with side information. IKRL (Xie et al., 2017) extends TransE by aligning visual features with entity embeddings, TransAE (Wang et al., 2019) integrates structural and visual signals through auto-encoding, RSME Wang et al. (2021) introduces a gating mechanism to filter noisy modalities, MKGformer (Chen et al., 2022) employs transformer-based fusion, and OTKGE (Cao et al., 2022) utilizes optimal transport to align heterogeneous information sources. Recent works such as MoSE (Zhao et al., 2022) and MoCoKGC (Li et al., 2024b) explore multi-source integration and momentum contrast, while LMKGE (Liu et al., 2024b) and LMKGE$_{l2}$ leverage Lorentzian geometry with contrastive learning to enhance semantic alignment.

### A.3.3 OUT-OF-DOMAIN INTENT DETECTION: DATASETS AND IMPLEMENTATION DETAILS

**Datasets.** Two highly-competitive OOD detection benchmarks are employed, including **BANKING** and **StackOverflow**. The BANKING dataset consists of 13,083 utterances, related to BANKING services, and is annotated with 77 unique intents. The StackOverflow dataset, sourced from Kaggle.com, encompasses 20,000 technical questions categorized into 20 distinct intents, with each class containing 1,000 samples.

**Implementation.** We compare *SPACL* with two recent baselines for OOD intent detection. MOGB (Li et al., 2025) constructs adaptive granular-ball decision boundaries to separate IND and OOD intents, while TBOS (Zhou et al., 2023) employs a tailored scoring strategy to jointly optimize IND and OOD detection. The widely used `BERT(-base-uncased)` model (12-layer Transformer) is adopted as the base encoder. We fine-tune the model using standard hyperparameters commonly recommended in prior work (Li et al., 2025; Zhou et al., 2023). The AdamW optimizer is employed, with the learning rate searched over $\{1^{e-5}, 2^{e-5}, 5^{e-5}, 5^{e-6}\}$, the batch size over $\{16, 32\}$, and the number of training epochs over $\{30, 50\}$.

## A.4 COMPUTATIONAL COMPLEXITY OF *SPACL*

The computational complexity of **SPACL** is summarized as follows. Let $b$ denote the batch size, $M$ the number of augmented views per sample, $d$ the feature dimension, and $\lambda_{\mathcal{P}^h}$ the number of hard positives per anchor. For each individual sample, we require:

(1) **Positive selection**: the time complexity is $\mathcal{O}(M^2 d + \lambda_{\mathcal{P}^h}^2 M)$, where the first term corresponds to constructing its similarity matrix and the second term comes from the greedy ranking. The overall complexity approximates $\mathcal{O}(d)$ as $d \gg M > \lambda_{\mathcal{P}^h}$.

(2) **Adversarial generation**: the time complexity is $\mathcal{O}\big((\lambda_{\mathcal{P}^h}+1)d^2 + (\lambda_{\mathcal{P}^h}+1)d\big)$, where the first term corresponds to the MLP (generator) of $\lambda_{\mathcal{P}^h}+1$ inputs and the second term comes from the second MLP (discriminator). The overall complexity approximates to $\mathcal{O}(d^2)$.

(3) **Negative screening**: the time complexity is $\mathcal{O}(bd + b\log b)$, where the first term corresponds to computing similarities between the anchor and all negatives, and the second term corresponds to the score ranking, thereby reducing the complexity approximately to $\mathcal{O}(bd)$.

Notably, all similarity scores (of anchor–positive and anchor–negative) have already been computed in steps (1) and (3). As a result, the final complexity is $\mathcal{O}(bd + bd^2 + b^2 d)$, given the batch size $b$.

We then compare *SPACL* with traditional CL methods, such as MoCo v1 (He et al., 2020), MoCo v3 (Chen et al., 2021), SimCLR (Chen et al., 2020a), and SimSiam (Chen & He, 2021), as shown in Table 4, where $K$ is the queue size for MoCo v1. Empirically, we also list all fine-tuning time on CIFAR-100 for 200 epochs using an NVIDIA A100 GPU server.

Table 4: Comparison of computational complexity and training time.

| Complexity source | Complexity | Training time |
|---|---|---|
| MoCo v1 | $\mathcal{O}(bKd + bK + bd)$ | 4 hours 45 minutes |
| MoCo v3 | $\mathcal{O}(b^2 d + bd^2 + bd + d^2)$ | 3 hours 51 minutes |
| SimCLR | $\mathcal{O}(b^2 d + bd^2 + bd + b^2)$ | 4 hours 03 minutes |
| SimSiam | $\mathcal{O}(bd^2 + bd)$ | 2 hours 46 minutes |
| *SPACL* | $\mathcal{O}(bd + bd^2 + b^2 d)$ | 3 hours 26 minutes |

We notice that the adversarial generation module is the main contributor to the additional computation. Nevertheless, the overall computational load of our method is comparable to that of MoCo v1/v3 and SimCLR, and our empirical results support this claim. In terms of memory usage, the additional components of *SPACL* compared to MoCo v1 are two MLPs: a generator and a discriminator, both implemented as one-hidden-layer MLPs. As a result, the additional memory usage is mainly determined by the number of hidden neurons (say $V$) as $\mathcal{O}(2Vd + Vd) = \mathcal{O}(3Vd)$.

## A.5 GENERALIZATION TO DIFFERENT BACKBONES

To address the backbone generality of *SPACL*, we further evaluate *SPACL* by employing ViT-S/16 and ViT-B/16 on ImageNet-1K. We strictly follow the setting from MoCo v3 (Chen et al., 2021), keeping the data augmentations and learning rate schedule unchanged, and simply substitute *SPACL* for the contrastive loss. For fairness, we also include SimCLR (Chen et al., 2020a), BYOL (Grill et al., 2020), and SwAV (Caron et al., 2020) under the same ViT training setup.

Table 5: Performance comparison between *SPACL* with other CL methods using ViT-based encoders. Note that $(*)$ denotes results are directly sourced from original papers, and the best resutls are **bold**.

| Model | MoCo v3 | SimCLR | BYOL | SwAV | *SPACL* |
|---|---|---|---|---|---|
| ViT-S, 300-ep | 72.5* | 69.0* | 71.0* | 67.1* | **74.3±0.12** |
| ViT-B, 300-ep | 76.5* | 73.9* | 73.9* | 71.6* | **78.6±0.10** |

As shown in Table 5, *SPACL* achieves the highest accuracy across both ViT-S/16 and ViT-B/16 encoders, outperforming other baselines. These results clearly confirm that the benefits of *SPACL* are not limited to convolutional backbones, but also exhibit strong performance on transformer-style encoders.

To incorporate more baselines, we also add DINO (Caron et al., 2021) as well as MAE (He et al., 2022), U-MAE (Zhang et al., 2022b), and i-MAE (Zhang & Shen, 2024) to our comparison, and report the results in Table 6. In this set of experiments, we use a ViT-B/16 encoder for all four methods and evaluate them on CIFAR-10, CIFAR-100, ImageNet-100, and ImageNet-1K.

Table 6: Performance comparison between *SPACL* and DINO/MAE-based methods. Note that ($\ast$) denotes results directly sourced from original papers, and the best results are in **bold**.

| Method | CIFAR-10 | CIFAR-100 | ImageNet-100 | ImageNet-1K |
|---|---|---|---|---|
| DINO | 89.52 | 66.76 | 74.84 | 78.2* |
| MAE | 90.78* | 68.66* | 72.23 | 67.48 |
| U-MAE | 68.90* | 61.42 | 67.50* | 58.50* |
| i-MAE | 92.00* | 69.50* | 74.63 | 70.21 |
| *SPACL* (ours) | **93.94±0.05** | **74.72±0.07** | **83.46±0.08** | **78.6±0.10** |

Across all four benchmarks, *SPACL* achieves the highest average performance, clearly surpassing both DINO and all MAE-based variants. For example, on ImageNet-100, the best-performing baseline is DINO with 74.84%. In contrast, *SPACL* attains 83.46%, outperforming DINO by a significant margin of +8.62% points, demonstrating a markedly stronger ability to learn discriminative representations. In addition, *SPACL* also demonstrates greater robustness across datasets of varying resolution and scale, showing that its design generalizes effectively from small-scale images (CIFAR-10/100) to mid-scale (ImageNet-100) and large-scale corpora (ImageNet-1K). Overall, *SPACL* consistently outperforms both DINO and the MAE-based methods, demonstrating superior performance.

## A.6 IMPACT OF POSITIVE AND NEGATIVE SAMPLE BUDGET

We examine the effect of sampling budgets for hard examples within *SPACL*. Specifically, we focus on two tunable hyperparameters: the *positive sample budget* $\lambda_{\mathcal{P}^h}$, which controls how many hard positives are selected from the augmented pool $\mathcal{P}_i$, and the *negative sample budget* $\lambda_{\mathcal{Q}^h}$, which determines what fraction of the combined in-batch and adversarial negatives are retained as hard negatives based on similarity-aware screening. To ensure a fair evaluation, we perform controlled ablations by varying one parameter at a time while fixing the other. That is, when analyzing the impact of $\lambda_{\mathcal{P}^h}$, we fix $\lambda_{\mathcal{Q}^h}$ at 95%. Conversely, when varying $\lambda_{\mathcal{Q}^h}$, we fix $\lambda_{\mathcal{P}^h}$ to 2 per anchor. This setup allows us to isolate the effect of each factor independently and provides a clear understanding of their roles in shaping the overall performance.

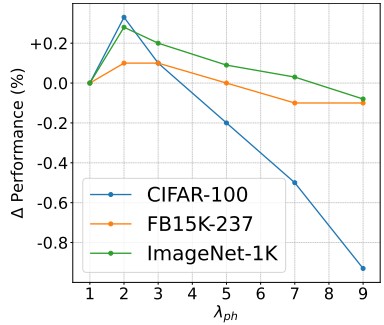
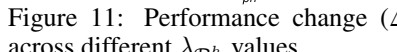
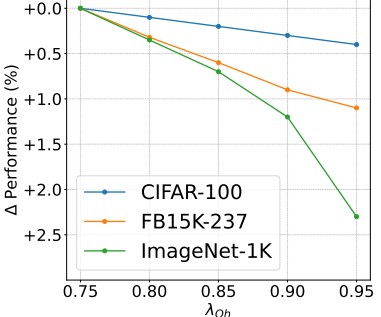

Figure 11: Performance change ($\Delta$) across different $\lambda_{\mathcal{P}^h}$ values.

Figure 12: Performance change ($\Delta$) across different $\lambda_{\mathcal{Q}^h}$ values.

The aggregated result presented in Fig. 11 reveals a notable trend in screening positives within the context of *SPACL* with the supervised learning paradigm. Specifically, as the increase of $\lambda_{\mathcal{P}^h}$, there is a corresponding decline in performance, with the highest performance achieved with $\lambda_{\mathcal{P}^h} = 2$ (for CIFAR-100) or $\lambda_{\mathcal{P}^h} = 3$ (for FB15K-237). One reason is that there are minimum constraints on the optimum with a small $\lambda_{\mathcal{P}^h}$ compared to other cases, that effectively increases the model learning capacity. Another reason could be attributed to Eq. (3), where a larger $\lambda_{\mathcal{P}^h}$ leads to a larger $s^+$ and decreases the norm of the gradient. Consequently, the gradient-based optimizer, commonly used in deep learning, may struggle to efficiently update model parameters, thereby compromising performance. However, relying solely on a single hard sample poses the risk of solution collapse, as indicated by **Theorem** 2.1. Consequently, we opt for a small $\lambda_{\mathcal{P}^h}$ with the value of 2 to mitigate

this risk. In addition, the comparison results for negatives are presented in Fig. 12. Notably, models trained with smaller values of $\lambda_{\mathcal{Q}^h}$ underperform compared to those using larger values, indicating that insufficient negative samples lead to weaker performance. Conversely, with all negatives (equivalent to $\lambda_{\mathcal{Q}^h} = 100\%$), performance drops substantially relative to the $90\%$ setting (as shown in Table 1), highlighting the importance of a *balanced* negative set. At last, we also notice that, ImageNet-1K is less sensitive to $\lambda_{Q^h}$, likely because of its higher intra-batch diversity, *i.e.*, a higher proportion of samples from different classes, reduces the probability that the additional samples selected by larger $\lambda$ values are trivial.