# OpenReview forum: "SNAPHARD CONTRAST LEARNING"
_ICLR.cc/2026/Conference — ICLR 2026 Poster_

### Official Review · Reviewer_MXc1 · 2025-10-18

**Soundness:** 3
**Presentation:** 4
**Contribution:** 3
**Rating:** 6
**Confidence:** 3

**Summary:**

This paper proposes a screening-based contrastive learning method to jointly sample hard positive and hard negative samples in contrastive learning for better representation diversity (via positives) and discriminability (via negatives). It also provides a theoretical analysis of sampling contrastive pairs and conditions for representation collapse.

**Strengths:**

1. Introduces a novel contrastive learning approach that jointly samples hard positives and negatives using a screening mechanism, with theoretical analysis supporting its design.
2. Shows strong empirical performance in 4 image classification benchmarks across and compares its method to supervised, self-supervised, and weakly-supervised baseline methods. Also compares its method to link prediction baselines in 3 benchmarks.
3. Provides a comprehensive ablation to show the effectiveness of the proposed method

**Weaknesses:**

1. In the supervised contrastive learning setting, the risk of false negatives is minimal since positives and negatives are explicitly defined by labels. However, in the self-supervised setting, false negatives can naturally occur when semantically similar samples are mistakenly treated as negatives. As SPACL’s focus on hard negatives can amplify the detrimental effect of false negatives, a discussion of false negatives would be beneficial.
2. I understand that “relying solely on a single hard sample poses the risk of solution collapse, as indicated by Theorem 2.1”. However, Figure 4 shows that increasing the number of hard positives beyond 4 leads to lower performance. Could the authors explain why increasing the number of hard positives (which are intended to prevent collapse) can lead to worse performance compared to using only one hard sample?

**Questions:**

1. What does the error bar in Table1, Table 2 indicate?
2. Please address questions in the weaknesses section.

---

> ### Author Response · Authors · 2025-11-25
> **Response to Reviewer MXc1**
>
> We are genuinely grateful for the reviewer's detailed comments and accurate understanding of our work. Below, we present a point-by-point response to your comments and outline the revisions made to address them.
>
>
> >**W1** In the supervised contrastive learning setting, the risk of false negatives is minimal since positives and negatives are explicitly defined by labels. However, in the self-supervised setting, false negatives can naturally occur when semantically similar samples are mistakenly treated as negatives. As SPACL’s focus on hard negatives can amplify the detrimental effect of false negatives, a discussion of false negatives would be beneficial.
>
>
> We appreciate the reviewer's thoughtful comment regarding false negatives. Conceptually, many false negatives are not truly semantic negatives; rather, they are typically semantically-related samples but happen to exhibit low similarity to the anchor. In this sense, *false negatives* are often similar to *hard positives* than genuine negatives.
>
> SPACL mitigates this issue in two complementary ways. First, our **hard-positive construction** explicitly selects semantically distant positive samples from the augmentation pool, thereby incorporating instances that would otherwise be treated as false negatives in in-batch CL. Second, unlike standard in-batch sampling (where “non-positive = negative”), SPACL applies a **similarity-based screening** for negatives. That is, only samples with **highest similarity to the anchor** are retained as hard negatives, while semantically unrelated or low-similarity instances, such as those potential false negatives, are **filtered out** before calculating the contrastive loss. This substantially reduces the likelihood that a false negative affects gradient updates.
>
> We again thank the reviewer for this thoughtful comment.
>
>
>
> >**W2** I understand that “relying solely on a single hard sample poses the risk of solution collapse, as indicated by Theorem 2.1”. However, Figure 4 shows that increasing the number of hard positives beyond 4 leads to lower performance. Could the authors explain why increasing the number of hard positives (which are intended to prevent collapse) can lead to worse performance compared to using only one hard sample?
>
>
> Thank you for this thoughtful question. As discussed in *Lines 463–468*, *Theorem 2.1* states that with single **one** hard positive, i.e., $\lambda_{\mathcal P^h}=1$, the anchor may collapse under the symmetry condition. Increasing $\lambda_{\mathcal P^h}$ from 1 to a *small set* (e.g., 2 or 3) effectively enlarges $\operatorname{conv}(\mathcal P^h)$ and breaks the symmetry, thereby preventing collapse.
>
> However, as shown in Eq. (3), increasing $\lambda_{\mathcal P^h}$ also increases the positive score $s^{+}$ (or $\sum_j \exp\langle x, p_j\rangle$), which in turn could reduce the norm of the gradient. This weaker gradient may restrict the encoder's ability to move away from fixation points, leading to over-regularizing the optimization. As such, this explains the trend in `Fig. 4`(`Fig. 6` in the revised manuscript`), where a few hard positives are sufficient to provide geometric stability, whereas too many can also hinder effective optimization.
>
>
>
>
> >**Q1** What does the error bar in Table 1, Table 2 indicate?
>
> The error bars reported in `Table 1/2` correspond to the standard deviation computed over five independent runs. We thank the reviewer for pointing this out and appreciate the opportunity to clarify this detail.

---

> > ### Comment · Reviewer_MXc1 · 2025-11-28
> >
> > Thanks for the author's detailed response. However, I have some concerns remaining about W1.
> >
> > **"That is, only samples with highest similarity to the anchor are retained as hard negatives, while semantically unrelated or low-similarity instances, such as those potential false negatives, are filtered out before calculating the contrastive loss. This substantially reduces the likelihood that a false negative affects gradient updates."**
> >
> > I think this interpretation appears inconsistent with the standard definition of false negatives in the contrastive learning literature [1,2]. Prior work consistently characterizes false negatives as high-similarity samples that are semantically related to the anchor but are mistakenly treated as negatives in the self-supervised setting. In fact, the core challenge identified in these works is precisely that false negatives and hard negatives both lie in the high-similarity region, making them difficult to distinguish.
> >
> > Therefore, filtering out low-similarity negatives does not address the false-negative issue. Instead, high-similarity negatives (which include both genuine hard negatives and potential false negatives) remain and continue to influence the gradient. For this reason, the explanation provided does not fully resolve the concern about how SPACL mitigates the detrimental effects of false negatives.
> >
> > That said, I appreciate the paper’s methodological contributions, and I will maintain my score.
> >
> > [1] Byun, Jaeseok, Dohoon Kim, and Taesup Moon. "Mafa: Managing false negatives for vision-language pre-training." Proceedings of the IEEE/CVF Conference on Computer Vision and Pattern Recognition. 2024.
> >
> > [2] Balmaseda, Vicente, et al. "Discovering Global False Negatives On the Fly for Self-supervised Contrastive Learning." arXiv preprint arXiv:2502.20612 (2025).

---

> > > ### Author Response · Authors · 2025-11-30
> > > **Response to Reviewer MXc1**
> > >
> > > We thank the reviewer again for the insightful clarification.
> > >
> > > We fully agree that false negatives are inherently difficult to separate from genuine hard negatives in self-supervised contrastive learning. Importantly, our adversarially generated negatives help construct stronger hard negatives than those found in in-batch sampling, including potential false negatives, as also shown in `Fig. 4` and `Fig. 5`. Because these adversarial negatives exhibit higher similarity to the anchor, the proposed selection mechanism naturally prioritizes them, thereby reducing the likelihood that moderately similar samples (which may include those false negatives) are selected as hard negatives. We apologise for the confusion caused by our previous wording: when we referred to *lower similarity*, we meant lower relative to the adversarially generated negatives.
> > >
> > > We also emphasise, as the reviewer correctly noted, that false negatives cannot be completely removed in self-supervised contrastive framework. While SPACL does not attempt to eliminate them, it reduces their relative influence by effectively filtering samples that fall outside the top-similarity region dominated by adversarial hard negatives. We have added this discussion in the revision (`Line 473-480`) and identify more explicit false-negative handling as an important direction for future work. We sincerely appreciate the reviewer’s valuable comments, which have significantly improved the quality of our work.

---

### Official Review · Reviewer_Qn56 · 2025-10-25

**Soundness:** 3
**Presentation:** 2
**Contribution:** 3
**Rating:** 6
**Confidence:** 3

**Summary:**

This paper introduces SnaPhArd Contrastive Learning (SPACL), which focuses on hard positive and negative examples within the contrastive learning (CL) framework. The authors analyzed the optimality and crash risks of CL and propose a method designed to overcome these challenges by using carefully selected contrasting pairs. SPACL demonstrates empirical performance by outperforming existing SOTA methods on multiple downstream tasks, including image classification and knowledge graph link prediction.

**Strengths:**

- The theoretical analysis provides a unique perspective for contrastive learning.
- Extensive experiments demonstrate the effectiveness of the method in this paper.
- The writing is clear and well-structured.

**Weaknesses:**

- This paper presents a detailed theoretical analysis; however, its heavy reliance on numerous equations makes the ideas somewhat obscure. The core concepts may be difficult for non-specialist readers in the field to grasp. Adding a few intuitive illustrations might enhance the overall completeness of the paper.

- While the paper demonstrates the efficacy of SPACL, the computational cost compared to other methods is not discussed in depth. Understanding the trade-offs in terms of computational requirements could be valuable, especially for real-world applications.

- I noticed that the experiments in this paper were conducted on classic datasets such as CIFAR and ImageNet. I understand that these are the standard datasets used in previous contrastive learning works. However, I have a more exploratory question: given the rapid and relatively mature development of VLMs (including large models across various domains), can the proposed method in this paper truly provide a valuable tool for the academic or industrial community?

**Questions:**

This paper verifies the effectiveness of SPACL on image classification and knowledge graph completion tasks. What challenges might SPACL encounter when dealing with other modalities or tasks? What are the advantages and disadvantages of SPACL when applied to different types of data or tasks?

---

> ### Author Response · Authors · 2025-11-25
> **[Part 1/3] Response to Reviewer Qn56**
>
> >**W1** This paper presents a detailed theoretical analysis; however, its heavy reliance on numerous equations makes the ideas somewhat obscure. The core concepts may be difficult for non-specialist readers in the field to grasp. Adding a few intuitive illustrations might enhance the overall completeness of the paper.
>
> We appreciate the reviewer's feedback regarding our theoretical analysis. Our goal in `Section 2` is to formally characterize the geometric conditions under which hard positives and negatives fundamentally contribute to the loss optimization. We acknowledge that some of them require familiarity with the geometry in CL formulations; therefore, we have included **intuitive geometric illustrations (`Fig. 8–11` in `Appendix A.1`)** to visually convey the core ideas for non-specialist readers.
> Importantly, `Section 3` then presents SPACL, with hard-sample screening strategy, which is built upon the theoretical analysis. Thus, users may apply SPACL without engaging with every theoretical detail, yet, the analysis remains essential for understanding the underlying rationale.
>
> >**W2** While the paper demonstrates the efficacy of SPACL, the computational cost compared to other methods is not discussed in depth. Understanding the trade-offs in terms of computational requirements could be valuable, especially for real-world applications.
>
>
> The computational complexity of **SPACL** is summarized as follows. Let $b$ denote the batch size, $M$ the number of augmented views per sample, $d$ the feature dimension, and $\lambda_{\mathcal P^h}$ the number of hard positives per anchor. *For each individual sample, we require*: (1) **Positive selection**: the time complexity is $\mathcal{O}(M^2d + \lambda_{\mathcal P^h}^2M)$, where the first term corresponds to constructing its similarity matrix and the second term comes from the greedy ranking. The overall complexity approximates $\mathcal{O}(d)$ as $d \gg M>\lambda_{\mathcal P^h}$. (2) **Adversarial generation**: the time complexity is $\mathcal{O}\big((\lambda_{\mathcal P^h}+1)d^2 + (\lambda_{\mathcal P^h}+1)d\big)$, where the first term corresponds to the MLP (generator) of $\lambda_{\mathcal P^h}+1$ inputs and the second term comes from the second MLP (discriminator). The overall complexity approximates to $\mathcal{O}(d^2)$. (3) **Negative screening**: the time complexity is $\mathcal{O}(bd+b\log b)$, where the first term corresponds to computing similarities between the anchor and all negatives, and the second term corresponds to the score ranking, thereby reducing the complexity approximately to $\mathcal{O}(bd)$. Notably,  all similarity scores (of anchor-positive and anchor-negative) have already been computed in step (1) and (3). As a result, the final  complexity is $\mathcal{O}(bd+bd^2 + b^2d)$, given the batch size of $b$.
>
>
> We then compare SPACL with traditional CL methods, such as MoCo v1[1], MoCo v3[2], SimCLR[3], SimSiam[4], as shown in **Table A**, where $K$ is the queue size for MoCo v1. Emprically, we also list all model's fine-tuning time on CIFAR-100 for 200 epochs using an NVIDIA A100 GPU server.
>
>
> **Table A. Comparison of computational complexity and training time.**
> | **Complexity source** | **Complexity** | **Training time** |
> |-----------------------|-------------------------|-------------------|
> | **MoCo v1[1]**           | $\mathcal{O}(bKd+bK+bd)$          | 4 hours 45 minutes|
> | **MoCo v3[2]**           | $\mathcal{O}(b^2d+bd^2+bd+d^2)$   | 3 hours 51 minutes|
> | **SimCLR[3]**            | $\mathcal{O}(b^2d+bd^2+bd+b^2)$   | 4 hours 03 minutes|
> | **SimSiam[4]**           | $\mathcal{O}(bd^2+bd)$            | 2 hours 46 minutes|
> | **SPACL**                | $\mathcal{O}(bd+bd^2 + b^2d)$  | 3 hours 26 minutes|
>
>
> We notice that the adversarial generation module is the main contributor to the additional computation. Nevertheless, the overall computational load of our method is comparable to that of MoCo v1/3 and SimCLR, and our empirical results support this claim. These points have been included in the revised manuscript under `Appendix A.4`. We sincerely thank the reviewer for this suggestion.
>
>
> [1] Momentum Contrast for Unsupervised Visual Representation Learning, CVPR 2020.
>
> [2] An Empirical Study of Training Self-Supervised Vision Transformers, ICCV 2021.
>
> [3] A Simple Framework for Contrastive Learning of Visual Representations, ICML 2020.
>
> [4] Exploring Simple Siamese Representation Learning, CVPR 2021.

---

> ### Author Response · Authors · 2025-11-25
> **[Part 2/3] Response to Reviewer Qn56**
>
> >**W3** The experiments in this paper were conducted on classic datasets such as CIFAR and ImageNet. While these are standard datasets for contrastive learning, given the rapid development of vision-language models (VLMs) and large models across domains, it would be worth exploring whether the proposed method could still provide academic or industrial value under these new paradigms.
>
>
> We appreciate this thoughtful suggestion. We agree that exploring SPACL with VLMs is valuable. We would like to emphasize that SPACL is a **self-supervised contrastive learning method** that is **agnostic to backbone architecture and downstream task**. Its core contribution is the **hard positive/negative screening**, which acts as a plug-and-play component that can be seamlessly integrated with either unimodal or multimodal encoders.
>
> For instance, `Section 4.2` already highlights the generality of SPACL in a **multimodal setting**, where each entity is associated with both visual and textual modalities. Additionally, we further evaluate SPACL with **ViT-S/16** and **ViT-B/16** (as shown in **Table B**) and observe consistent improvements, demonstrating that SPACL transfers effectively to transformer-based encoders commonly used in VLMs.
>
>
> **Table B: Performance comparison between SPACL wiht other CL methods using ViT-based encoders.Note that ( * ) denotes results are directly sourced from original papers, and the best resutls are bold.**
> | Model | MoCo v3[1] | SimCLR[2] | BYOL[3] | SwAV[4] | **SPACL** |
> |----------------|---------|--------|------|------|-----------|
> | ViT-S, 300-ep | 72.5(*) | 69.0(*) | 71.0(*) | 67.1(*) | **74.3±0.12** |
> |ViT-B, 300-ep|76.5(*)|73.9(*)|73.9(*)|71.6(*)|**78.6±0.10**|
>
> Given these observations, we believe SPACL naturally extends to VLMs, where integration would primarily involve **sample argumentation, selecting hard positives, generating adversarial negatives, and replacing the standard InfoNCE loss with our hard-sample screening variant**, without modification to the underlying backbone. We will explore this promising direction in the future work. We appreciate the reviewer’s valuable suggestion and these points have been included in the revised manuscript under `Appendix A.5`.

---

> ### Author Response · Authors · 2025-11-25
> **[Part 3/3] Response to Reviewer Qn56**
>
> >**Q1** This paper verifies the effectiveness of SPACL on image classification and knowledge graph completion tasks. What challenges might SPACL encounter when dealing with other modalities or tasks? What are the advantages and disadvantages of SPACL when applied to different types of data or tasks?
>
>
> We appreciate the reviewer's thoughtful question regarding data types or tasks. Our current experiments focus on focus on vision (`Section 4.1`) and multimodalitiy reasoning (`Section 4.2`). To further strengthen the evaluation, we additionally include results from a purely text-domain setting.
>
> Specifically, we introduce an out-of-domain (OOD) intent detection task, a standard benchmark to evaluate both (i) classification performance on in-domain (IND) intents and (ii) results when encountering out-of-domain (OOD) intents. Following prior work [1–2], we adopt the widely used BANKING and StackOverflow benchmarks, and vary the ratio of IND classes used during training (25%, 50%, 75%). For evaluation, we report macro F1 on out-of-domain intents (F1-OOD) and macro F1 on in-domain intents (F1-IND).
>
>
> **Table C: OOD detection performances across the BANKING and StackOverflow datasets under different proportions of IND classes (25%, 50%, 75%). Note that ( * ) denotes results are directly sourced from original papers, and the best resutls are bold.**
> | Ratio | Methods     | BANKING F1-OOD | BANKING F1-IND | StackOverflow F1-OOD | StackOverflow F1-IND |
> |-------|-------------|----------------|----------------|----------------------|----------------------|
> | 25%   | MOGB[1]  | 88.29(*)          | 74.50(*)          | 94.42(*)               | **87.80(*)**                |
> |       | TBOS[2]     | 92.70 (*)         | 79.70 (*)           | 95.52 (*)                 | 85.35 (*)                 |
> |       | SPACL   | **93.51±0.05**     | **80.21±0.09**     | **96.72±0.06**           | 86.24±0.14           |
> | 50%   | MOGB[1]  | **89.71(*)**          | **87.27(*)**          | 81.04(*)                | 81.53(*)                |
> |       | TBOS[2]     | 84.42          | 84.63          | 90.41                | 88.96                |
> |       | SPACL   | 85.48±0.03     | 85.57±0.18     | **91.52±0.19**           | **90.12±0.08**           |
> | 75%   | MOGB[1]  | 75.52(*)          | 88.16(*)          | 71.27(*)                | 87.09(*)                |
> |       | TBOS[2]     | 74.94 (*)           | 89.23 (*)           | 75.58 (*)                 | 88.40 (*)                 |
> |       | SPACL   | **75.72±0.02**     | **89.95±0.16**     | **76.39±0.18**           | **88.96±0.05**           |
>
> As shown in **Table C**, SPACL achieves competitive performance with recent baselines across two datasets, achieving the best in-domain accuracy and OOD detection performance on average. These results demonstrate that the proposed SPACL is modality-agnostic and generalizes effectively across purely visual/textual, and multimodal CL settings. We also acknowledge that *modality-specific augmentation and distribution-alignment techniques* may be necessary for optimal performance. We appreciate the reviewer’s valuable suggestion and these points have been included in the revised manuscript under `Section 4.3`.
>
>
>
> [1] Multi-Granularity Open Intent Classification via Adaptive Granular-Ball Decision Boundary, AAAI 2025.
>
> [2] Two Birds One Stone: Dynamic Ensemble for OOD Intent Classification, ACL 2023.

---

### Official Review · Reviewer_PoEY · 2025-10-25

**Soundness:** 3
**Presentation:** 3
**Contribution:** 2
**Rating:** 4
**Confidence:** 3

**Summary:**

This paper investigates the role of hard sample mining in Contrastive Learning (CL). The authors provide a theoretical analysis of the InfoNCE loss, deriving optimality conditions to motivate the importance of hard samples. Based on this analysis, they propose SnaPhArd Contrast Learning (SPACL), a method that systematically screens for and utilizes hard positive and hard negative samples. The paper reports strong empirical results on image classification and knowledge graph link prediction tasks, showing improvements over a selected set of baseline methods.

**Strengths:**

- **Clear Theoretical Grounding**: A major strength is the clean and elegant theoretical motivation derived from the InfoNCE loss function. This provides a principled "why" for the algorithm's design, which is commendable.
- **Thorough Ablation Studies**: The paper includes an extensive set of ablation experiments that methodically dissect the SPACL algorithm. This rigorous analysis convincingly demonstrates the utility of each proposed component within its framework.

**Weaknesses:**

- **Limited Scope of Experimental Comparison**: The evaluation could be strengthened by including a broader set of baselines. Comparisons are notably missing against: (a) the current state-of-the-art paradigm of Masked Image Modeling (e.g., MAE), and (b) influential non-InfoNCE contrastive methods like DINO. This makes it difficult to assess the method's standing in the wider field of self-supervised learning.
- **Lack of Discussion on Complexity**: The proposed method adds several components (e.g., an adversarial generator and a discriminator) that increase complexity. The paper would benefit from a discussion on the trade-off between this added complexity and the resulting performance gains.

**Questions:**

- For clarity and reproducibility, we suggest moving key experimental details, such as the backbone architecture, from the appendix to the main paper.
- Could you please clarify the computational overhead (e.g., in terms of training time or GPU memory) introduced by the SPACL components, particularly the adversarial generator and discriminator, compared to a simpler baseline like SimCLR?
- It is recommended to provide more details on the usage of LLMs in research ideation/writing.

---

> ### Author Response · Authors · 2025-11-25
> **[Part 1/2] Response to Reviewer PoEY**
>
> We sincerely appreciate the reviewer's insightful comments. We have thoroughly reviewed each point and incorporated the necessary revisions and clarifications to address the concerns:
>
> > **W1** Limited Scope of Experimental Comparison:The evaluation could be strengthened by including a broader set of baselines. Comparisons are notably missing against: (a) the current state-of-the-art paradigm of Masked Image Modeling (e.g., MAE), and (b) influential non-InfoNCE contrastive methods like DINO. This makes it difficult to assess the method’s standing in the wider field of self-supervised learning.
>
>
> We sincerely thank the reviewer for highlighting the importance of including more baselines. As suggested, we have incorporated DINO [1], together with MAE [2], U-MAE [3] and i-MAE [4]. On CIFAR-10, CIFAR-100, ImageNet-100, and ImageNet-1K image-classification benchmarks, we further compare SPACL with these baselines, in which the ViT-B/16 encoder is employed.
>
>
>
>
> **Table A. Performance comparison between SPACL and other DINO and MAE based methods. Note that (*) denotes results are directly sourced from original papers, the best resutls are bold.**
> | Method           | CIFAR-10 | CIFAR-100 | ImageNet-100 | ImageNet-1K |
> | ---------------- | -------- | --------- | ------------ | -------- |
> |DINO[1]            |89.52|66.76|74.84|78.2(*) |
> | MAE[2]             | 90.78(*) | 68.66(*) | 72.23 | 67.48 |
> | U-MAE[3]         | 68.90(*) | 61.42 | 67.50(*) | 58.50(*) |
> | i-MAE [4]         | 92.00(*) | 69.50(*) | 74.63 | 70.21 |
> | **SPACL（ours）** | **93.94±0.05** | **74.72±0.07** | **83.46±0.08** | **78.6±0.10** |
>
> Across all benchmarks, SPACL consistently outperforms both DINO and MAE-based methods, demonstrating superior accuracy. We thank the reviewer for this constructive suggestion. The inclusion of these additional baselines broadens the empirical evaluation and provides a clearer understanding of our method.
>
>
> [1] Emerging Properties in Self-Supervised Vision Transformers, ICCV 2021.
>
> [2] Masked Autoencoders Are Scalable Vision Learners, CVPR 2022.
>
> [3] How Mask Matters: Towards Theoretical Understandings of Masked Autoencoders, NeurIPS 2022.
>
> [4] i-MAE: Are Latent Representations in Masked Autoencoders Linearly Separable?, CVPR Workshops 2024.

---

> ### Author Response · Authors · 2025-11-25
> **[Part 2/2] Response to Reviewer PoEY**
>
> >**W2/Q2** Lack of Discussion on Complexity: The proposed method adds several components (e.g., an adversarial generator and a discriminator) that increase complexity. The paper would benefit from a discussion on the trade-off between this added complexity and the resulting performance gains.||Could you please clarify the computational overhead (e.g., in terms of training time or GPU memory) introduced by the SPACL components, particularly the adversarial generator and discriminator, compared to a simpler baseline like SimCLR?
>
>
> The computational complexity of **SPACL** is summarized as follows. Let $b$ denote the batch size, $M$ the number of augmented views per sample, $d$ the feature dimension, and $\lambda_{\mathcal P^h}$ the number of hard positives per anchor. *For each individual sample, we require*: (1) **Positive selection**: the time complexity is $\mathcal{O}(M^2d + \lambda_{\mathcal P^h}^2M)$, where the first term corresponds to constructing its similarity matrix and the second term comes from the greedy ranking. The overall complexity approximates $\mathcal{O}(d)$ as $d \gg M>\lambda_{\mathcal P^h}$. (2) **Adversarial generation**: the time complexity is $\mathcal{O}\big((\lambda_{\mathcal P^h}+1)d^2 + (\lambda_{\mathcal P^h}+1)d\big)$, where the first term corresponds to the MLP (generator) of $\lambda_{\mathcal P^h}+1$ inputs and the second term comes from the second MLP (discriminator). The overall complexity approximates to $\mathcal{O}(d^2)$. (3) **Negative screening**: the time complexity is $\mathcal{O}(bd+b\log b)$, where the first term corresponds to computing similarities between the anchor and all negatives, and the second term corresponds to the score ranking, thereby reducing the complexity approximately to $\mathcal{O}(bd)$. Notably,  all similarity scores (of anchor-positive and anchor-negative) have already been computed in step (1) and (3). As a result, the final  complexity is $\mathcal{O}(bd+bd^2 + b^2d)$, given the batch size of $b$.
>
>
> We then compare SPACL with traditional CL methods, such as MoCo v1[1], MoCo v3[2], SimCLR[3], SimSiam[4], as shown in **Table B**, where $K$ is the queue size for MoCo v1. Emprically, we also list all model's fine-tuning time on CIFAR-100 for 200 epochs using an NVIDIA A100 GPU server.
>
>
> **Table B. Comparison of computational complexity and training time.**
> | **Complexity source** | **Complexity** | **Training time** |
> |-----------------------|-------------------------|-------------------|
> | **MoCo v1[1]**           | $\mathcal{O}(bKd+bK+bd)$          | 4 hours 45 minutes|
> | **MoCo v3[2]**           | $\mathcal{O}(b^2d+bd^2+bd+d^2)$   | 3 hours 51 minutes|
> | **SimCLR[3]**            | $\mathcal{O}(b^2d+bd^2+bd+b^2)$   | 4 hours 03 minutes|
> | **SimSiam[4]**           | $\mathcal{O}(bd^2+bd)$            | 2 hours 46 minutes|
> | **SPACL**                | $\mathcal{O}(bd+bd^2 + b^2d)$  | 3 hours 26 minutes|
>
>
> We notice that the adversarial generation module is the main contributor to the additional computation. Nevertheless, the overall computational load of our method is comparable to that of MoCo v1/3 and SimCLR, and our empirical results support this claim. These points have been included in the revised manuscript under `Appendix A.4`. We sincerely thank the reviewer for this suggestion.
>
>
> [1] Momentum Contrast for Unsupervised Visual Representation Learning, CVPR 2020.
>
> [2] An Empirical Study of Training Self-Supervised Vision Transformers, ICCV 2021.
>
> [3] A Simple Framework for Contrastive Learning of Visual Representations, ICML 2020.
>
> [4] Exploring Simple Siamese Representation Learning, CVPR 2021.
>
> >**Q1** For clarity and reproducibility, we suggest moving key experimental details, such as the backbone architecture, from the appendix to the main paper.
>
> We appreciate the reviewer's suggestion regarding clarity and reproducibility. In the revision, we move the key experimental details, including benchmark descriptions and backbone specifications, from the appendix into the main paper. Thank you for this helpful suggestion.
>
>
> >**Q3** It is recommended to provide more details on the usage of LLMs in research ideation/writing.
>
> We would like to clarify that no LLMs are used for research ideation or writing this manuscript.

---

> ### Comment · Reviewer_PoEY · 2025-11-26
> **Official Comments by Reviewer PoEY**
>
> Thanks for the authors' detailed response. Since most of my concerns have been addressed, I have raised my rating to 6.

---

> > ### Author Response · Authors · 2025-11-26
> >
> > Dear Reviewer PoEY
> >
> > We are glad that our response addresses your concerns. Thank you for your helpful and valuable questions. We truly appreciate any additional thoughts you might have on our work.

---

### Official Review · Reviewer_tz8p · 2025-10-25

**Soundness:** 3
**Presentation:** 3
**Contribution:** 3
**Rating:** 6
**Confidence:** 2

**Summary:**

Contrastive learning has attracted considerable attention in recent years and has demonstrated strong empirical performance across various tasks. To address the issue of potential false negative samples in the construction of feature pairs, this paper conducts a comprehensive theoretical analysis from the perspective of optimality conditions and proposes the SPACL framework. Specifically, the method improves the process of constructing positive and negative pairs by selecting “hard” positive and “hard” negative samples, rather than treating all samples equally. The experimental results show that the proposed approach achieves excellent performance.

**Strengths:**

S1: This paper provides a relatively thorough theoretical analysis, outlining the optimality conditions and highlighting the beneficial role of hard samples in constructing feature pairs. The proposed geometric analysis offers a novel perspective and is compelling.

S2: At the methodological level, SPACL integrates the selection of distant positive samples with adversarially generated hard negative samples, which effectively enhances the discriminative capability of the model and serves as a valuable complement to existing methods.

S3: The extensive experimental results clearly validate the effectiveness of the proposed method, showing consistent and significant improvements over other methods across various benchmark datasets.

S4: The paper provides a number of illustrative figures, which are helpful for understanding the proposed method.

**Weaknesses:**

W1: Intuitively, the proposed method could incur some additional computational cost. Thus, providing an analysis of memory usage and throughput would help to more comprehensively evaluate the efficiency of the method.

W2: In the ablation study, the authors state that the performance drop in the “w/o AN” setting indicates that adversarially generated negative samples help sharpen the boundary of the negative region; otherwise, relying solely on batch-based or queue-based negatives would lead to a more blurred boundary. It would be helpful to provide a more fine-grained explanation or additional visualizations to further support this claim.

**Questions:**

Please refer to the weaknesses section. The main issues are concentrated on the performance analysis and experiments, as well as the need for further explanation or visualization in the ablation studies.

---

> ### Author Response · Authors · 2025-11-25
> **[Part 1/2] Response to Reviewer tz8p**
>
> We sincerely thank the reviewer for the constructive feedback and valuable insights. We have carefully revised the manuscript to address these comments and incorporate the suggested improvements. Our detailed point-by-point response is provided below:
>
> > **W1** Intuitively, the proposed method could incur some additional computational cost. Thus, providing an analysis of memory usage and throughput would help to more comprehensively evaluate the efficiency of the method.
>
>
> The computational complexity of **SPACL** is summarized as follows. Let $b$ denote the batch size, $M$ the number of augmented views per sample, $d$ the feature dimension, and $\lambda_{\mathcal P^h}$ the number of hard positives per anchor. *For each individual sample, we require*: (1) **Positive selection**: the time complexity is $\mathcal{O}(M^2d + \lambda_{\mathcal P^h}^2M)$, where the first term corresponds to constructing its similarity matrix and the second term comes from the greedy ranking. The overall complexity approximates $\mathcal{O}(d)$ as $d \gg M>\lambda_{\mathcal P^h}$. (2) **Adversarial generation**: the time complexity is $\mathcal{O}\big((\lambda_{\mathcal P^h}+1)d^2 + (\lambda_{\mathcal P^h}+1)d\big)$, where the first term corresponds to the MLP (generator) of $\lambda_{\mathcal P^h}+1$ inputs and the second term comes from the second MLP (discriminator). The overall complexity approximates to $\mathcal{O}(d^2)$. (3) **Negative screening**: the time complexity is $\mathcal{O}(bd+b\log b)$, where the first term corresponds to computing similarities between the anchor and all negatives, and the second term corresponds to the score ranking, thereby reducing the complexity approximately to $\mathcal{O}(bd)$. Notably,  all similarity scores (of anchor-positive and anchor-negative) have already been computed in step (1) and (3). As a result, the final  complexity is $\mathcal{O}(bd+bd^2 + b^2d)$, given the batch size of $b$.
>
>
> We then compare SPACL with traditional CL methods, such as MoCo v1[1], MoCo v3[2], SimCLR[3], SimSiam[4], as shown in **Table A**, where $K$ is the queue size for MoCo v1. Emprically, we also list all model's fine-tuning time on CIFAR-100 for 200 epochs using an NVIDIA A100 GPU server.
>
>
> **Table A. Comparison of computational complexity and training time.**
> | **Complexity source** | **Complexity** | **Training time** |
> |-----------------------|-------------------------|-------------------|
> | **MoCo v1[1]**           | $\mathcal{O}(bKd+bK+bd)$          | 4 hours 45 minutes|
> | **MoCo v3[2]**           | $\mathcal{O}(b^2d+bd^2+bd+d^2)$   | 3 hours 51 minutes|
> | **SimCLR[3]**            | $\mathcal{O}(b^2d+bd^2+bd+b^2)$   | 4 hours 03 minutes|
> | **SimSiam[4]**           | $\mathcal{O}(bd^2+bd)$            | 2 hours 46 minutes|
> | **SPACL**                | $\mathcal{O}(bd+bd^2 + b^2d)$  | 3 hours 26 minutes|
>
>
> We notice that the adversarial generation module is the main contributor to the additional computation. Nevertheless, the overall computational load of our method is comparable to that of MoCo v1/3 and SimCLR, and our empirical results support this claim. In terms of the memory usage, the additional components of SPACL compared to MoCo v1 are two MLPs: a generator and a discriminator. Both of them are implemented as a one-hidden-layer MLP. As a result, the additional memory usage is mainly determined by the number of hidden neuron (say $V$) as $\mathcal{O}(2Vd+Vd)=\mathcal{O}(3Vd)$. These points have been included in the revised manuscript under `Appendix A.4`. We sincerely thank the reviewer for this suggestion.
>
>
> [1] Momentum Contrast for Unsupervised Visual Representation Learning, CVPR 2020.
>
> [2] An Empirical Study of Training Self-Supervised Vision Transformers, ICCV 2021.
>
> [3] A Simple Framework for Contrastive Learning of Visual Representations, ICML 2020.
>
> [4] Exploring Simple Siamese Representation Learning, CVPR 2021.

---

> > ### Author Response · Authors · 2025-11-25
> > **[Part 2/2] Response to Reviewer tz8p**
> >
> > > **W2** In the ablation study, the authors state that the performance drop in the “w/o AN” setting indicates that adversarially generated negative samples help sharpen the boundary of the negative region; otherwise, relying solely on batch-based or queue-based negatives would lead to a more blurred boundary. It would be helpful to provide a more fine-grained explanation or additional visualizations to further support this claim.
> >
> > We thank the reviewer for this helpful suggestion. To further illustrate the effect of our adversarial negatives, we sample CIFAR-100 with a batch size of 256, randomly select one example as the anchor, generate three negatives using our adversarial module, and visualize all embeddings via t-SNE. In this visualization, most in-batch negatives remain in distant clusters, whereas the adversarial negatives concentrate in the region surrounding the anchor. This result is consistent with our analysis from (`Line 430`), that is, adversarial negatives effectively tighten the local decision boundary and enhance the model. These points have been included in the revised manuscript under `Section 4.3.3`.

---

> > > ### Comment · Reviewer_tz8p · 2025-11-26
> > >
> > > Thank you for your response. Some of my concerns have been addressed, so I will keep my original score.

---

> ### Author Response · Authors · 2025-11-26
>
> Dear Reviewer tz8p,
>
> We are pleased that our responses have addressed your concerns. Should you have any additional questions or suggestions, we would be glad to continue the discussion.

---

### Official Review · Reviewer_61L9 · 2025-11-01

**Soundness:** 4
**Presentation:** 3
**Contribution:** 4
**Rating:** 6
**Confidence:** 4

**Summary:**

The paper presents a theoretical and empirical investigation into contrastive learning (CL) with a focus on hard positive and negative sample selection. The authors first analyze the optimality and collapse conditions of CL through geometric and gradient-based formulations of the InfoNCE loss. Building upon this theoretical framework, they propose SPACL, a new algorithm that strategically emphasizes “hard” samples - instances that are challenging for the model to distinguish, while filtering out trivial ones.

SPACL introduces two core innovations: 1) Hard Positive Selection: Identifies the most diverse and dissimilar augmented views via a farthest-point iterative process, thereby mitigating representation collapse. 2) Hard Negative Generation and Screening: Incorporates adversarially generated negatives and a relative screening mechanism to retain only the most informative (hard) negative samples.

Empirical results across multiple benchmarks, including CIFAR-10/100, ImageNet(-100), and three knowledge graph datasets (WN9, FB15K-237, FB15K), show consistent improvements over state-of-the-art methods such as SimCLR, MoCo, SupCon, and VarCon. Theoretical derivations are complemented by extensive ablation studies demonstrating the contribution of each module.

**Strengths:**

1. This paper provides complete analyses of why and how sample hardness influences contrastive optimization, connecting gradient geometry and convex hull dynamics.

2. The dual hard-sample selection (positive and negative) is elegantly motivated and empirically validated.


3. Strong performance across multiple benchmarks and supervision levels, with consistent margins of improvement.


4. Clearly isolate the role of each component (anchor selection, hard positives, adversarial negatives, and screening).


5. Demonstrates applicability in both visual and relational (knowledge graph) contexts.

**Weaknesses:**

1. The derivations assume idealized settings (e.g., constant auxiliary encoder ggg, unit-sphere normalization) that may not hold in practical large-scale CL systems. The paper could better discuss these approximations.


2. While effective, SPACL introduces additional overhead via adversarial negative generation and iterative positive selection. The paper lacks a quantitative runtime or resource comparison to baselines.


3. Although two distinct domains are tested, the method’s performance in text or multimodal CL remains unexamined.


4. Although λ parameters are ablated, the dependence of SPACL’s stability on these values may warrant further analysis, especially for large-scale datasets.

**Questions:**

1. Could the authors provide a more explicit analysis of SPACL’s computational and memory complexity relative to MoCo or SupCon? How feasible is adversarial negative generation in large-scale pretraining?


2. Have the authors tested SPACL with transformer-based encoders (e.g., ViT, CLIP) to verify whether the observed benefits generalize beyond convolutional backbones?


3. The hardness definition for positives depends on dissimilarity averaging. Have alternative formulations (e.g., mutual information or gradient-based difficulty) been explored?


4. Since the adversarial component introduces an inner min–max loop, did the authors encounter stability issues during training? If so, how were they mitigated?


5. The conclusion mentions potential applicability to negative-free CL frameworks (e.g., BYOL, SimSiam). Could the authors clarify how the proposed geometric analysis might translate to such setups?

---

> ### Author Response · Authors · 2025-11-25
> **[Part 1/5] Response to Reviewer 61L9**
>
> Thank you for your thorough review and insightful comments on our manuscript. We have carefully considered your feedback and have made the following revisions and clarifications to address the raised concerns.
>
> > **W1** The derivations assume idealized settings (e.g., constant auxiliary encoder g, unit-sphere normalization) that may not hold in practical large-scale CL systems. The paper could better discuss these approximations.
>
> We thank the reviewer for raising this valuable point. Our derivations aligns with the standard optimization paradigm when there are multiple blocks of model parameters, such as  ADMM, where _one block is fixed_ while optmizing **the other to its optimal**. This is reflected in our analysis where the auxiliary encoder $g$ is treated as constant (fixing its parameters) during optimization and the optimality of $f$ is explored to obtain clean geometric characterizations of InfoNCE-based contrastive learning (see `Eq. (3)`, `Theorem 2.1` and `Corollary 2.2`). Notably, MoCo-style CL frameworks inherently rely on this alternating optimization as well: one encoder is updated while the momentum encoder remains constant within a short time window. The unit-sphere constraint is also necessary so that representation direction determines contrastive similarity without the interference of the scales/lengths of the vectors.
>
> To clarify, our theoretical results concern the shape of the optimization landscape and the impact of hard/easy samples, rather than dataset size or backbone encoders. The analysis characterizes how the optimal representation $x$ behaves under constant $g$, and this reasoning naturally generalizes to large-scale CL settings. To empirically validate this claim, we also evaluate SPACL on ImageNet-1K, a widely recognized large-scale benchmark (see `Table 1` in our manuscript). SPACL consistently outperforms other baselines, demonstrating its robustness and effectiveness even at large scale.

---

> ### Author Response · Authors · 2025-11-25
> **[Part 2/5] Response to Reviewer 61L9**
>
> > **W2&Q1** While effective, SPACL introduces additional overhead via adversarial negative generation and iterative positive selection. The paper lacks a quantitative runtime or resource comparison to baselines. Could the authors provide a more explicit analysis of SPACL’s computational and memory complexity relative to MoCo or SupCon? How feasible is adversarial negative generation in large-scale pretraining?
>
> The computational complexity of **SPACL** is summarized as follows. Let $b$ denote the batch size, $M$ the number of augmented views per sample, $d$ the feature dimension, and $\lambda_{\mathcal P^h}$ the number of hard positives per anchor. *For each individual sample, we require*: (1) **Positive selection**: the time complexity is $\mathcal{O}(M^2d + \lambda_{\mathcal P^h}^2M)$, where the first term corresponds to constructing its similarity matrix and the second term comes from the greedy ranking. The overall complexity approximates $\mathcal{O}(d)$ as $d \gg M>\lambda_{\mathcal P^h}$. (2) **Adversarial generation**: the time complexity is $\mathcal{O}\big((\lambda_{\mathcal P^h}+1)d^2 + (\lambda_{\mathcal P^h}+1)d\big)$, where the first term corresponds to the MLP (generator) of $\lambda_{\mathcal P^h}+1$ inputs and the second term comes from the second MLP (discriminator). The overall complexity approximates to $\mathcal{O}(d^2)$. (3) **Negative screening**: the time complexity is $\mathcal{O}(bd+b\log b)$, where the first term corresponds to computing similarities between the anchor and all negatives, and the second term corresponds to the score ranking, thereby reducing the complexity approximately to $\mathcal{O}(bd)$. Notably,  all similarity scores (of anchor-positive and anchor-negative) have already been computed in step (1) and (3). As a result, the final  complexity is $\mathcal{O}(bd+bd^2 + b^2d)$, given the batch size of $b$.
>
>
> We then compare SPACL with traditional CL methods, such as MoCo v1[1], MoCo v3[2], SimCLR[3], SimSiam[4], as shown in **Table A**, where $K$ is the queue size for MoCo v1. Emprically, we also list all model's fine-tuning time on CIFAR-100 for 200 epochs using an NVIDIA A100 GPU server.
>
>
> **Table A. Comparison of computational complexity and training time.**
> | **Complexity source** | **Complexity** | **Training time** |
> |-----------------------|-------------------------|-------------------|
> | **MoCo v1[1]**           | $\mathcal{O}(bKd+bK+bd)$          | 4 hours 45 minutes|
> | **MoCo v3[2]**           | $\mathcal{O}(b^2d+bd^2+bd+d^2)$   | 3 hours 51 minutes|
> | **SimCLR[3]**            | $\mathcal{O}(b^2d+bd^2+bd+b^2)$   | 4 hours 03 minutes|
> | **SimSiam[4]**           | $\mathcal{O}(bd^2+bd)$            | 2 hours 46 minutes|
> | **SPACL**                | $\mathcal{O}(bd+bd^2 + b^2d)$  | 3 hours 26 minutes|
>
>
> We notice that the adversarial generation module is the main contributor to the additional computation. Nevertheless, the overall computational load of our method is comparable to that of MoCo v1/3 and SimCLR, and our empirical results support this claim. These points have been included in the revised manuscript under `Appendix A.4`. We sincerely thank the reviewer for this suggestion.
>
>
> [1] Momentum Contrast for Unsupervised Visual Representation Learning, CVPR 2020.
>
> [2] An Empirical Study of Training Self-Supervised Vision Transformers, ICCV 2021.
>
> [3] A Simple Framework for Contrastive Learning of Visual Representations, ICML 2020.
>
> [4] Exploring Simple Siamese Representation Learning, CVPR 2021.

---

> ### Author Response · Authors · 2025-11-25
> **[Part 3/5] Response to Reviewer 61L9**
>
> > **W3** Although two distinct domains are tested, the method's performance in text or multimodal CL remains unexamined.
>
> We thank the reviewer for this constructive suggestion. It indeed motivates a more comprehensive examination of SPACL across different modalities. We would like to clarify that our current experimental setup *already spans distinct modalities*. Specifically, `Section 4.1` evaluates SPACL on visual setting, while `Section 4.2` evaluates SPACL on a **multimodal setting**, as each entity is represented using both symbolic triples and visual features (see `Appendix A.3.2`, `Lines 1158`, `1162`, `1171`). We acknowledge that this multimodal aspect may not have been sufficiently highlighted in the main text, which may have contributed to the reviewer's impression.
>
>
> To further strengthen the evaluation, we additionally include purely text-domain experiments. Specifically, we introduce an out-of-domain (OOD) intent detection task, a standard benchmark to evaluate both (i) classification performance on in-domain (IND) intents and (ii) results when encountering out-of-domain (OOD) intents. Following prior work [1–2], we adopt the widely used BANKING and StackOverflow benchmarks, and vary the ratio of IND classes used during training (25%, 50%, 75%). For evaluation, we report macro F1 on out-of-domain intents (F1-OOD) and macro F1 on in-domain intents (F1-IND).
>
>
> **Table B: OOD detection performances across the BANKING and StackOverflow datasets under different proportions of IND classes (25%, 50%, 75%). Note that ( * ) denotes results are directly sourced from original papers, and the best resutls are bold.**
> | Ratio | Methods     | BANKING F1-OOD | BANKING F1-IND | StackOverflow F1-OOD | StackOverflow F1-IND |
> |-------|-------------|----------------|----------------|----------------------|----------------------|
> | 25%   | MOGB[1]  | 88.29(*)          | 74.50(*)          | 94.42(*)               | **87.80(*)**                |
> |       | TBOS[2]     | 92.70 (*)         | 79.70 (*)           | 95.52 (*)                 | 85.35 (*)                 |
> |       | SPACL   | **93.51±0.05**     | **80.21±0.09**     | **96.72±0.06**           | 86.24±0.14           |
> | 50%   | MOGB[1]  | **89.71(*)**          | **87.27(*)**          | 81.04(*)                | 81.53(*)                |
> |       | TBOS[2]     | 84.42          | 84.63          | 90.41                | 88.96                |
> |       | SPACL   | 85.48±0.03     | 85.57±0.18     | **91.52±0.19**           | **90.12±0.08**           |
> | 75%   | MOGB[1]  | 75.52(*)          | 88.16(*)          | 71.27(*)                | 87.09(*)                |
> |       | TBOS[2]     | 74.94 (*)           | 89.23 (*)           | 75.58 (*)                 | 88.40 (*)                 |
> |       | SPACL   | **75.72±0.02**     | **89.95±0.16**     | **76.39±0.18**           | **88.96±0.05**           |
>
> As shown in **Table B**, SPACL achieves competitive performance with recent baselines across two datasets, achieving the best in-domain accuracy and OOD detection performance on average. These results demonstrate that the proposed SPACL is modality-agnostic and generalizes effectively across purely visual/textual, and multimodal CL settings. We appreciate the reviewer’s valuable suggestion and these points have been included in the revised manuscript under `Section 4.3`.
>
> [1] Multi-Granularity Open Intent Classification via Adaptive Granular-Ball Decision Boundary, AAAI 2025.
>
> [2] Two Birds One Stone: Dynamic Ensemble for OOD Intent Classification, ACL 2023.
>
> > **W4** Although $\lambda$ parameters are ablated, the dependence of SPACL’s stability on these values may warrant further analysis, especially for large-scale datasets.
>
> We thank the reviewer for this constructive and insightful comment, as $\lambda_{P^h}$ and $\lambda_{Q^h}$ play an important role in SPACL. Motivated by this suggestion, we extend our study by performing an additional *fully supervised* ablation on ImageNet-1K, and we have updated `Fig. 6/7` accordingly to present these results.
>
> Across all datasets, including ImageNet-1K, we observe that a larger $\lambda_{P^h}$ leads to performance degradation. This aligns with our theoretical motivation (as discussed in `Lines 463–465`). On the other hand, the performance peaks when $\lambda_{Q^h}=0.95$, as noted from `Lines 470–472`. Notably, ImageNet-1K is less sensitive to $\lambda_{P^h}$ or $\lambda_{Q^h}$, likely because of its higher intra-batch diversity (i.e., a higher proportion of samples from different classes) reduces the probability that the additional samples selected by larger $\lambda$ values are trivial.
>
> We appreciate the reviewer for prompting this valuable analysis.

---

> > ### Author Response · Authors · 2025-11-25
> > **[Part 4/5] Response to Reviewer 61L9**
> >
> > > **Q2** Have the authors tested SPACL with transformer-based encoders (e.g., ViT, CLIP) to verify whether the observed benefits generalize beyond convolutional backbones?
> >
> > Thank you for raising this important question regarding the backbone generality of SPACL. As noted in `Line 1151`, we intentionally use ResNet(-50) as a unified backbone for all experiments in `Section 4.1` to isolate the effect of backbones and enable apples-to-apples comparison with only CL settings.
> >
> > To address the reviewer's concern, we further evaluate SPACL by employing ViT-S/16 and ViT-B/16 on ImageNet-1K. We strictly follow the setting from MoCo v3 [1], keeping the data augmentations and learning rate schedule unchanged, and simply substitute SPACL for the contrastive loss. For fairness, we also include SimCLR [2], BYOL [3], and SwAV [4] under the same ViT training setup.
> >
> > **Table C: Performance comparison between SPACL wiht other CL methods using ViT-based encoders.Note that ( * ) denotes results are directly sourced from original papers, and the best resutls are bold.**
> > | Model | MoCo v3[1] | SimCLR[2] | BYOL[3] | SwAV[4] | **SPACL** |
> > |----------------|---------|--------|------|------|-----------|
> > | ViT-S, 300-ep | 72.5(*) | 69.0(*) | 71.0(*) | 67.1(*) | **74.3±0.12** |
> > |ViT-B, 300-ep|76.5(*)|73.9(*)|73.9(*)|71.6(*)|**78.6±0.10**|
> >
> > As shown in **Table C**, SPACL achieves the highest accuracy across both ViT-S/16 and ViT-B/16 encoders, outperforming other baselines. These results clearly confirm that the benefits of SPACL are not limited to convolutional backbones, but also effectively to transformer-style encoders. We appreciate the reviewer’s valuable suggestion and these points have been included in the revised manuscript under `Appendix A.5`.
> >
> > [1] An Empirical Study of Training Self-Supervised Vision Transformers, ICCV 2021.
> >
> > [2] A Simple Framework for Contrastive Learning of Visual Representations, ICML 2020.
> >
> > [3] Bootstrap Your Own Latent, NeurIPS 2020.
> >
> > [4] Unsupervised Learning of Visual Features by Contrasting Cluster Assignments, NeurIPS 2020.
> >
> > > **Q3** The hardness definition for positives depends on dissimilarity averaging. Have alternative formulations (e.g., mutual information or gradient-based difficulty) been explored?
> >
> > We appreciate this constructive suggestion regarding the choice of hardness definition. In our current setting, the hardness of positive candidates is measured using inner-product–based dissimilarity. This choice is deliberate: the inner product is the core similarity used in the contrastive loss (`Eq. (9)`); as such, aligning the hardness criterion with the contrastive objective ensures consistency in model optimization. Notably, cosine-based dissimilarity can also be used here as inner product and cosine similarity differ only by $l_2$-normalization.
> >
> > We agree that alternatives, such as mutual-information or gradient based difficulty, are promising and potentially complementary directions. We view these alternatives as valuable extensions, and plan to explore them in future work.
> >
> > > **Q4** Since the adversarial component introduces an inner–max loop, did the authors encounter stability issues during training? If so, how were they mitigated?
> >
> > We thank the reviewer for raising this question regarding potential instability caused by the adversarial training. Yet, in practice, we did not observe any training instability. To verify, we specifically visualize both the generator and the discriminator loss on a CIFAR-100 task.
> >
> > The observation is summarized as below. The discriminator loss exhibits a smooth, gradually increasing trend, while the generator loss decreases throughout training. We attribute this result to the following reason: early in training, the generator might produce weaker adversarial examples, making them easy for the discriminator to classify, thereby resulting in high generator loss and low discriminator loss. As training progresses, the generator becomes stronger and produces harder samples, causing the discriminator loss to increase, while the generator loss continues to decrease as it adapts to the discriminator. Importantly, both components exhibit stability and are optimized smoothly. These points have been included in the revised manuscript under `Section 4.3.3`.

---

> > > ### Author Response · Authors · 2025-11-25
> > > **[Part 5/5] Response to Reviewer 61L9**
> > >
> > > > **Q5** The conclusion mentions potential applicability to negative-free CL frameworks (e.g., BYOL, SimSiam). Could the authors clarify how the proposed geometric analysis might translate to such setups?
> > >
> > > We appreciate the reviewer's interest in the potential applicability to negative-free CL frameworks. Notably, these methods (including SPACL) share a structural similarity with dual-encoder learning: both employ two encoders, where one branch is optimized while the other is treated as fixed (i.e., stop-gradient in BYOL and SimSiam). In this positive-only case, our collapse condition (*Theorem 2.1*) reduces to analyzing whether the anchor can degenerate toward the target representation, and the same geometric reasoning could explain why negative-free methods require architectural asymmetry (e.g., stop-gradient) to avoid collapsing. We plan to apply this analysis to those cases to gain more understanding of them in our future research.

---

### Author Response · Authors · 2025-11-30
**Revised version has already uploaded**

Thank you to all reviewers for the thoughtful and constructive feedback. We are encouraged that the reviews recognize the theoretical contributions of our geometric analysis (`Reviewers 61L9, PoEY`), the empirical effectiveness of SPACL across multimodal tasks (`Reviewers tz8p, Qn56, MXc1`), and the clarity of the overall presentation (`Reviewers tz8p, MXc1`).

Your feedback has helped us significantly improve the manuscript. Following all received comments, we summarize the major revisions below and highlight them in the order of their appearance in the revised manuscript:


* **Clarifying existing multimodal evaluation `(Section 4.2)`:**
`Section 4.2` evaluates SPACL in a multimodal KG setting, where each entity is represented using both textual descriptions and visual images (see `Appendix A.3.2`, Lines `1250`, `1258`, `1270`). In the revision, we explicitly highlight this multimodal setup to strengthen readers’ understanding and avoid potential ambiguity.


* **New text-domain evaluation `(Section 4.3)`:**
We introduce a purely text-domain OOD intent-detection task (two benchmarks) to further demonstrate that SPACL generalizes beyond visual-only settings. The corresponding results are reported in `Table 3`.

* **Visualizations for adversarial training `(Figures 4–5)`:**
We add a t-SNE visualization comparing in-batch and adversarial negatives (`Figure 4`), along with training curves for both the adversarial generator and discriminator (`Figure 5`), to better illustrate the embedding behavior and the effect of adversarial hard-negative generation.

* **Hyperparameter analysis `(Figures 6–7)`:**
We update the analyses of $\lambda_{\mathcal P^h}$ and $\lambda_{\mathcal Q^h}$ with revised figures and expanded discussions, providing clearer insights into their influence, particularly on large-scale benchmarks.

* **Computational complexity analysis `(Appendix A.4)`:**
We add a new section that presents SPACL’s computational complexity and compares it with existing CL methods, including MoCo v1/v3, SimCLR, and SimSiam, in `Table 4`.

* **Generalization to different backbones `(Appendix A.5)`:**
To demonstrate that SPACL is backbone-agnostic, we add two ViT-based(ViT-S and ViT-B) experiments and report the results in `Table 5`, showing consistent improvements with transformer-style encoders. Also we incorporate four additional baselines base on ViT-B/16 encoder, i.e., DINO, MAE, U-MAE, and i-MAE, to provide a more comprehensive comparison and report the results in `Table 6`.

We sincerely appreciate the reviewers' constructive feedback and hope that the revised manuscript satisfactorily addresses all major concerns.

---

### Author Response · Authors · 2025-12-02
**Rebuttal Summary**

Dear AC, SAC, and Reviewers:

Thank you for your time and effort in reviewing our submission. To provide a comprehensive yet concise overview, we include a *brief summary* of *reviewer concerns*, *corresponding actions*, and *discussion outcomes* (presented in the order of the reviewers). We hope this helps you verify how each major concern has been discussed and resolved. More details can be found either in the individual replies to each reviewer or in our general response (`Revised version has already been uploaded` on 01 Dec 2025).


### * Reviewer `61L9`

**Concerns/Actions**: *(W1)* Provide geometric rationale and validation on the large-scale benchmark; *(W2 & Q1)* Present computational-complexity comparisons with mainstream methods; *(W3)* Clarify multimodal setup and include pure text-domain evaluations; *(W4)* Extend hyperparameter analysis on the large-scale dataset; *(Q2)* Add experiments with the transformer-based encoder (ViT); *(Q3)* Justify the use of inner-product hardness; *(Q4)* Provide evidence on training stability; *(Q5)* Discuss potential applicability to negative-free settings.

**Discussion outcome**: No further replies were posted after our responses (from **25 Nov, 2025**).



### * Reviewer `tz8p`

**Concerns/Actions**:
*(W1)* Present computational-complexity comparisons with mainstream methods; *(W2)* Add t-SNE visualizations showing the boundary separation of adversarial and in-batch negatives.

**Discussion outcome**: Reviewer explicitly replied **"I will keep my original score"** (**26 Nov 2025**), without further questions.


### * Reviewer `PoEY`

**Concerns/Actions**: (W1) Add experiments comparing with DINO- and MAE-based baselines; (W2 & Q2) Present computational-complexity comparisons with mainstream methods; (Q1) Clarify backbone and benchmark details in the main paper; (Q3) Confirm that no LLMs are used.

**Discussion outcome**: Reviewer explicitly replied **"I have raised my rating to 6"** (**26 Nov 2025**), indicating concerns were addressed.

### * Reviewer `Qn56`

**Concerns/Actions**: *(W1)* Clarify methodology and theoretical intuition; *(W2)* Present computational-complexity comparisons with mainstream methods; *(W3)* Explain extension to VLMs with ViT-based experiments; *(Q1)* Clarify multimodal setup and include pure text-domain evaluations.

**Discussion outcome**: No further reply was posted after our responses (from **25 Nov, 2025**).

### * Reviewer `MXc1`

**Concerns/Actions**: *(W1)* Explain how adversarial negatives help in false-negative problem; *(W2)* Clarify the model performance when using more positives in `Fig. 4`; *(Q1)* Explain the definition of the error bars; *(W1 Follow-up)*: Explain how adversarial negatives weaken the impact of false negatives by generating stronger hard positives.

**Discussion outcome**: The reviewer explicitly replied **“I appreciate the paper’s methodological contributions, and I will maintain my score”** (**28 Nov 2025**).


### * Final remark
Overall, the revised version, with all major changes highlighted, has been uploaded after addressing all concerns based on the latest discussion. We believe that our rebuttal and additional experiments have thoroughly addressed the reviewers’ questions. We sincerely thank all five reviewers for their constructive and insightful suggestions, which have greatly strengthened the manuscript. Please feel free to let us know if you have any further questions or suggestions.

Sincerely,

Authors

---

### Meta-Review · Area_Chair_G69b · 2026-01-04

**Summary:**

This paper proposes SPACL, a contrastive learning framework that systematically uses hard positive and hard negative samples. The authors provide a theoretical analysis of the InfoNCE objective to motivate the importance of hard samples. Reviewers agree that the paper is based on a clear and strong theoretical motivation. The proposed method is well designed and empirically validated, showing effectiveness across a wide range of benchmarks in both vision and knowledge graph domains. Reviewers also acknowledged the comprehensive ablation validating the contribution of each module. Overall, the paper makes a meaningful and well-supported contribution to contrastive learning, and I recommend accepting it.

**Reviewer Concerns:**

In the original reviews, reviewers raised questions about computational overhead, assumptions in the theoretical analysis, and broader applicability, these concerns were largely framed as requests for clarification and further discussion rather than fundamental flaws. Overall, reviewers expressed positive opinions of the soundness and contribution of the work in both theory and empirical practice.

**Reviewer Scores:**

Reviewer 61L9 and Qn56 were originally positive about the paper. Their questions were mainly about clarification of the method, complexity, and additional experiments. These were addressed in the rebuttal and I would expect that they'll remain positive.

Reviewer tz8p, PoEY, MXc1 have explicitly noted in their response that most concerns were addressed and they were positive toward the work.

---

### Decision · Program_Chairs · 2026-01-26

Accept (Poster)